# Control of a gene transfer agent cluster in *Caulobacter crescentus* by transcriptional activation and anti-termination

Ngat T. Tran[1] & Tung B. K. Le [1] ✉

Gene Transfer Agents (GTAs) are phage-like particles that cannot self-multiply and be infectious. *Caulobacter crescentus*, a bacterium best known as a model organism to study bacterial cell biology and cell cycle regulation, has recently been demonstrated to produce bona fide GTA particles (CcGTA). Since *C. crescentus* ultimately die to release GTA particles, the production of GTA particles must be tightly regulated and integrated with the host physiology to prevent a collapse in cell population. Two direct activators of the CcGTA biosynthetic gene cluster, GafY and GafZ, have been identified, however, it is unknown how GafYZ controls transcription or how they coordinate gene expression of the CcGTA gene cluster with other accessory genes elsewhere on the genome for complete CcGTA production. Here, we show that the CcGTA gene cluster is transcriptionally co-activated by GafY, integration host factor (IHF), and by GafZ-mediated transcription anti-termination. We present evidence that GafZ is a transcription anti-terminator that likely forms an anti-termination complex with RNA polymerase, NusA, NusG, and NusE to bypass transcription terminators within the 14 kb CcGTA cluster. Overall, we reveal a two-tier regulation that coordinates the synthesis of GTA particles in *C. crescentus*.

Viruses and mobile genetic elements are major drivers of evolution in bacteria[1,2]. In some cases, there is evidence that these elements might be co-opted to perform biological functions for the host, thus providing selective advantages for the producing organisms[3–7]. Virus-like Gene Transfer Agents (GTAs) are among these cases[3,4,8,9]. GTAs were first discovered and characterized in the non-sulfur photosynthetic bacterium *Rhodobacter capsulatus*[10], but have now been found in a wide range of bacterial species and archaea[3,4,11,12]. Homologs of GTA core genes are common in abundant environmental organisms, and it has been speculated that ~10^6 virus-like particles per milliliter of marine water could be GTAs[11,13–16]. GTAs are thought to be domesticated prophages that can no longer self-multiply and be infectious, but can still transfer DNA via a DNA-filled capsid head[3,17–22]. In contrast to canonical phages, which preferably package and transfer their own DNA, genomic DNA from the host is packaged into GTA particles in a relatively non-specific manner, although some packaging bias exists in certain species[3,4,20,23,24]. Most notably, the length of DNA packaged into GTA particles is not sufficient to contain the entire gene cluster encoding GTA components[3,22]. Moreover, multiple accessory genes necessary for GTA synthesis are distributed across the host genome, sometimes megabases away from the main GTA gene cluster[3,21,23,24], thus GTAs cannot transfer themselves horizontally nor be self-replicative and infectious.

Since host bacteria ultimately die to release GTA particles, the production of GTA particles must be tightly regulated and integrated with the host physiology to prevent a collapse in cell population. In *R. capsulatus*, RcGTA gene expression and production are under the control of multiple host regulatory systems including the CckA-ChpT-CtrA phosphorelay, quorum sensing, and a sigma/anti-sigma-like mechanism[25–35]. Recently, a direct and dedicated activator of the RcGTA gene cluster, GafA, was identified[36]. Further characterization of this multi-domain activator by Sherlock and Fogg (2022)

[1]Department of Molecular Microbiology, John Innes Centre, Norwich NR4 7UH, UK. ✉e-mail: tung.le@jic.ac.uk

demonstrated that GafA controls RcGTA gene expression via a direct interaction with the RNA polymerase omega subunit (RpoZ-ω), potentially recruiting RNA polymerase to GTA-specific promoters for transcriptional activation[37].

More recently, we and colleagues demonstrated that the α-proteobacterium *Caulobacter crescentus*, best known as a model organism to study cell cycle regulation[38], produces bona fide GTA particles (CcGTA)[23]. In this bacterium, a 14-kb long gene cluster, spanning from CCNA02880 (here named *gtaT*) to CCNA02861 (here named *gtaB*), encodes the majority of genes required for the synthesis of GTA particles[23] (Fig. 1a). CcGTAs were shown to encapsulate host genomic DNA (~8.3 kb on average)[23], most likely through head-full packaging[17]. Unlike RcGTA, CcGTA synthesis is repressed under normal laboratory conditions by the transcriptional repressor RogA[23] (Fig. 1a), but CcGTA particles are produced in a subpopulation of cells once *rogA* is removed genetically[23]. RogA exerts its repression by binding directly to the core promoter region of the *gafYZ* operon (Fig. 1a), the two direct activators of the CcGTA cluster[23]. Notably, *C. crescentus* GafY and GafZ show sequence similarity to the N-terminal and C-terminal domains of *R. capsulatus* GafA, respectively[12,23]. *C. crescentus* GafY and GafZ were suggested to co-regulate the promoter upstream of *gtaT* (Fig. 1a), the first gene of the CcGTA gene cluster, to transcriptionally activate the cluster[23]. However, it is unknown how GafYZ controls transcription mechanistically, and how they coordinate expression of the main CcGTA gene cluster with the accessory genes elsewhere on the genome for CcGTA production.

To investigate, we combined a genetic screen, genome-wide deep sequencing, and biochemical approaches, and demonstrate that the *C. crescentus* GTA gene cluster is transcriptionally co-activated by integration host factor (IHF) and by GafYZ-mediated transcription anti-termination. We show that, in the absence of the repressor RogA, IHF directly binds to the promoter region of the *gafYZ* operon, and together with GafY, co-activates *gafYZ* transcription. Furthermore, IHF also binds upstream of the CcGTA cluster, and together with GafY and GafZ, mediates transcription activation and anti-termination to express the entire CcGTA cluster. We present evidence that GafZ is a transcription anti-terminator that likely forms an anti-termination complex with RNA polymerase, NusA, NusG, and NusE to bypass transcription terminators identified within the CcGTA cluster. Overall, we reveal an exquisite two-tier regulation that coordinates the synthesis of GTA particles in *C. crescentus*.

## Results

### Activation of the GTA gene cluster requires the integration host factor IHF

Previously, Gozzi et al.[23] showed that the production of *C. crescentus* GTA is tightly regulated by a repressor, RogA, which represses the transcription of the *gafYZ* operon, encoding the direct activators of GTA synthesis[23] (Fig. 1a). To further understand the control of the GTA cluster, we devised a blue-white screen based on a *lacA* reporter to identify additional factors that might act together with or independently of GafYZ. We engineered a transcriptional fusion of *lacA* to *gtaM* (encoding the GTA major capsid gene) in a Δ*lacA* background (Fig. 1a). The resulting strain produced white colonies on agar media containing X-gal (Fig. 1b), consistent with the previous finding that the GTA cluster is transcriptionally silent under normal lab conditions[23]. As expected, deleting *rogA* alleviated transcription repression of the GTA cluster[23], producing blue colonies (Fig. 1b). We transposon-mutagenized this Δ*rogA* i.e., GTA-on indicator strain and looked for rare white colonies. Confirming that the screen worked, we found transposon insertions within *gafY* and in the promoter of *gafYZ*, the operon encoding the two known GTA activators[23] (Fig. 1b). In addition, we found a single insertion in the small (291-bp) *ihfB* gene, which encodes the β subunit of integration host factor (IHF) (Fig. 1b). Complementation of this strain

by expressing *ihfB* ectopically from the *xyl* locus restored LacA activity (Fig. 1b), suggesting that IHF is required to activate the GTA cluster. IHF is known to be a heterodimer, consisting of an α and a β subunit[39]. To investigate further, we deleted *ihfA* and *ihfB* individually from the GTA-on Δ*rogA* strain, then assayed for the presence of packaged GTA DNA and the head-tail connector protein (GtaL) as proxies for the synthesis of GTA particles (Fig. 1c, d). In the GTA-on Δ*rogA* strain, packaged GTA DNA can be seen as a distinct ~9-kb band in undigested total DNA samples[23] (Fig. 1c). Deletion of either *ihfA* or *ihfB* prevented detectable GTA DNA (Fig. 1c). Also, no detectable GtaL protein was produced in Δ*ihfA/B* Δ*rogA* strains, as assessed by an immunoblot using an anti-GtaL antibody (Fig. 1d). GTA production was restored when *ihfA/B* was expressed ectopically from the *xyl* locus (Fig. 1c, d). Altogether, our data show that IHF is required for GTA synthesis in *C. crescentus*.

### IHF binds the promoter regions of *gafYZ* and the GTA gene cluster to activate gene expression

IHF is a known DNA-binding and bending transcriptional factor[39–42], but its regulon has not been fully characterized in *C. crescentus*[43,44]. To further understand the roles IHF might play in GTA synthesis, we performed chromatin immunoprecipitation with deep sequencing (ChIP-seq) to map the genome-wide binding sites of FLAG-tagged IHFα and IHFβ in the GTA-off background (stationary phase WT *C. crescentus*) as well as the GTA-on background (stationary phase Δ*rogA C. crescentus*) (Supplementary Fig. 1a). Non-tagged WT and Δ*rogA* strains were employed as negative controls to eliminate false signals that might arise from cross-reaction with the anti-FLAG antibody (Supplementary Fig. 1a). ChIP-seq revealed (i) there are numerous enriched IHF binding sites on the *C. crescentus* chromosome (430 peaks with $\log_2$(fold enrichment) >2, $-\log_{10}$($q$ value) > 350), (ii) the ChIP-seq peaks of IHFα and IHFβ overlap, consistent with IHF functioning as an αβ heterodimer, and (iii) the binding of IHF to DNA is independent of RogA (Supplementary Fig. 1a). A closer inspection of anti-FLAG-IHF ChIP-seq datasets revealed a single peak in the promoter region of the *gafYZ* operon (Fig. 2a), and five peaks in the GTA gene cluster in both Δ*rogA* and WT backgrounds (one of which, IBE1, is in the promoter region of the GTA cluster) (Fig. 2b). This suggested that IHF might directly regulate transcription of *gafYZ* and of the GTA gene cluster, a possibility that we confirmed by quantitative reverse transcriptase PCR (qRT-PCR) (Fig. 2c). Deletion of *rogA* caused an upregulation of *gafY* by ~350-fold, and of *gtaT* by ~1500-fold, ultimately leading to the synthesis of GTA particles (Fig. 2c). When either *ihfA* or *ihfB* was deleted, the elevated gene expression of *gtaT* in the Δ*rogA* background was abolished, whereas that of *gafY* was reduced ~9-fold (Fig. 2c). The upregulation of *gtaT* and *gafY* in the Δ*rogA* background was restored when *ihfA* or *ihfB* was ectopically expressed from the *xyl* locus (Fig. 2c).

MEME analysis using the hundred most enriched sites in anti-FLAG-IHF ChIP-seq datasets allowed us to propose a consensus DNA-binding motif for *C. crescentus* IHF (Supplementary Fig. 1b), in turn allowing us to locate the IHF binding elements (IBEs) in the GTA promoter (IBE1) and the *gafYZ* promoter (IBE6) (Fig. 2a, b). We experimentally verified these IHF binding elements by replacing the conserved TT in the putative IHF binding site with GG, and then monitoring the binding of FLAG-tagged IHF by ChIP-seq. The TT → GG mutations selectively eliminated the enrichment of IHF at the DNA region under investigation (Fig. 2a, b), and nowhere else on the chromosome. When the IHF binding element in either the *gafYZ* promoter or the GTA promoter was mutated, GTA was no longer produced, as shown by the absence of both the packaged GTA DNA and GtaL protein (Fig. 2d, e).

We also observed four additional IHF binding elements (IBE 2-3-4-5) within the GTA cluster, in between the large terminase-encoding gene *gtaT* and the portal protein-encoding gene *gtaP* (Fig. 2b). To investigate the contribution of these sites to GTA

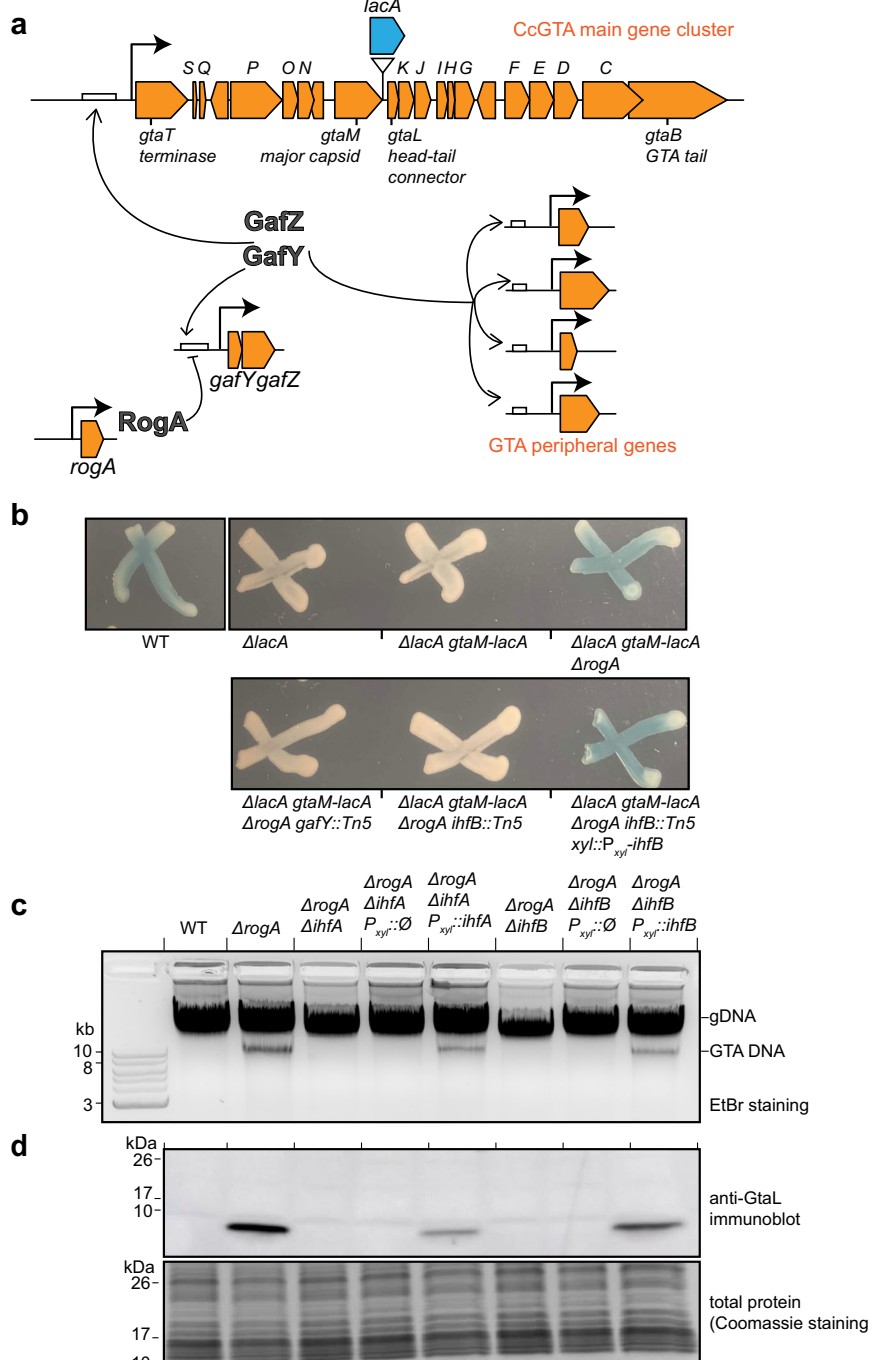

**Fig. 1 | The integration host factor (IHF) is required for GTA production.**
**a** Diagram of CcGTA regulation. RogA directly represses (flat-headed arrow) the expression of *gafYZ*. GafY and GafZ directly activate (arrows) the main GTA gene cluster (orange) and GTA peripheral genes (orange) on the *C. crescentus* chromosome. For the full list of main CcGTA cluster genes and their annotated functions, see Supplementary Data 1. For the Tn5-based screen for additional GTA regulators, *lacA* gene (blue block arrow) was transcriptionally fused downstream of the GTA major capsid gene *gtaM*. *C. crescentus* forms blue colonies on agar media supplemented with a chromogenic lactose analog (X-gal) owing to the presence of a membrane-bound dehydrogenase LacA that is necessary for converting lactose/lactose analog into molecules that can be imported into the cytoplasm for

subsequent hydrolysis[76], while *ΔlacA* colonies are white on such media (Fig. 1b).
**b** Phenotypes of indicated strains on agar media containing X-gal and 0.03% xylose.
**c** Total DNA extraction from indicated strains in stationary phase. Total DNA was purified and separated by electrophoresis on a 1% agarose gel which was stained with ethidium bromide (EtBr) for DNA. Experiments were performed at least twice, and a representative image is shown. **d** Immunoblot of total cell lysates of indicated strains using a polyclonal anti-GtaL (GTA head-tail connector protein) antibody. A separate Coomassie-stained SDS-PAGE was loaded with the same volume of samples to serve as a loading control. Immunoblots were performed at least twice, and a representative image is shown. Source data are provided as a Source data file.

synthesis, we eliminated them by making TT → GG/TG mutations while maintaining the coding sequence of the underlying *gtaQS* genes (Supplementary Fig. 2a). ChIP-seq of FLAG-IHF confirmed that such mutations indeed eliminated IHF binding to the four

elements internal to the GTA cluster (Supplementary Fig. 2a), however, the resulting strains were still able to produce GTA particles, as suggested by the presence of packaged GTA DNA and GtaL protein (Supplementary Fig. 2b, c). We concluded that these four

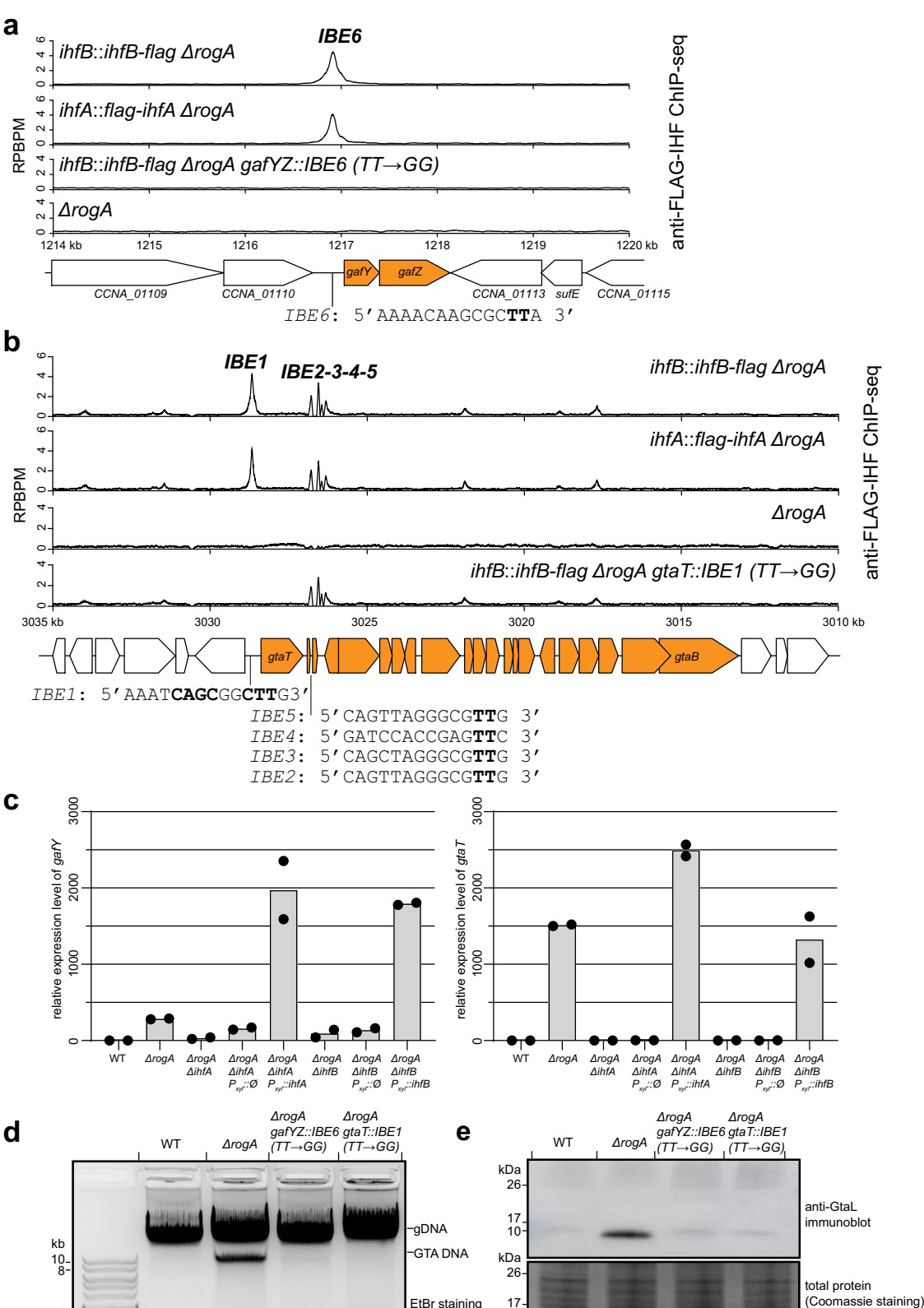

internal IHF binding sites play little or no role in the transcription activation of the GTA cluster. Altogether, our results suggest that IHF activates transcription of the GTA phage cluster by binding directly to the promoter of the GTA cluster, and to the promoter of the *gafYZ* operon, which encodes the two direct activators of the GTA cluster.

## IHF binds to GafY-regulated promoters

Our previous anti-GafY ChIP-seq showed that GafY binds 18 sites on the chromosome, in addition to the promoter regions of *gafYZ* and the GTA cluster itself [23]. The ChIP-seq datasets showed that 18 out of 20 GafY-binding DNA regions are also co-occupied by IHF (Supplementary Fig. 3). This suggested that IHF and GafY might work together to

**Fig. 2 | IHF binds the promoter regions of *gafYZ* and the GTA gene cluster to activate gene expression. a, b** anti-FLAG ChIP-seq profiles show the enrichment of FLAG-tagged IHFα and β in different genetic backgrounds. Profiles were plotted with the x-axis representing genomic positions and the y-axis representing the number of reads per base pair per million mapped reads (RPBPM). The positions and sequences of identified IHF-binding elements (IBE) are shown beneath the schematic diagram of genes (GTA-related genes are depicted as orange block arrows). The conserved TT nucleotides in IBEs were highlighted in bold, and were mutated to GG to eliminate IHF binding. Note that the GTA main gene cluster is on the minus strand, but was inverted to run from left to right for a presentation purpose only. ChIP-seq experiments were performed twice using biological replicates, and a representative profile is shown. MACS2-identified ChIP-seq peaks above IBE 1, 2, 3, and 6 are reproducible in both replicates and significant i.e., having Poisson distribution $-\log_{10}(p$ value) and false discovery rate $-\log_{10}(q$ value) > 1000 in both replicates. ChIP-seq peaks above IBE 4 and 5 were not reliably detected by MACS2 but have recognizable IHF-binding motif and were further characterized in Supplementary Fig. 2. **c** Relative gene expression of *gafY* (left panel) and *gtaT* (right panel) from indicated strains in stationary phase, as quantified by qRT-PCR ($n = 2$). **d** Total DNA extraction from indicated strains grown up to a stationary phase. Experiments were performed at least twice, and a representative image is shown. **e** Immunoblot of total cell lysates of indicated strains using a polyclonal anti-GtaL (GTA head-tail connector protein) antibody. A separate Coomassie-stained SDS-PAGE was loaded with the same volume of samples to serve as a loading control. Immunoblots were performed at least twice, and a representative image is shown. Source data are provided as a Source data file.

activate transcription, a possibility that we investigated using the promoter region of the GTA cluster.

## Multiple binding elements at the GTA cluster promoter region are required for transcriptional activation

To investigate how IHF and GafYZ might work together to activate the transcription of the GTA cluster, we used RNA-seq to locate the possible transcription start sites (TSS) of the GTA cluster in the GTA-on *ΔrogA* strain (Fig. 3a). We discovered a single TSS, labeled A + 1, in the upstream region of the GTA cluster (Fig. 3a). This identified TSS, however, lies 110 bp downstream of the computationally annotated start codon (TTG) of *gtaT*, suggesting mis-annotation (Fig. 3a). Indeed, mutating this TTG to CTG did not affect GTA synthesis, as judged by the presence of packaged GTA DNA in total DNA purification (Fig. 3b). Furthermore, deleting a DNA region ranging from +90 to +214, which would have rendered the remaining downstream region of *gtaT* out of frame (if TTG was a correct start codon), did not affect GTA production either (Fig. 3b). Downstream of *gtaT* TSS, we identified three possible in-frame ATG start codons (Fig. 3a). Individually mutating these candidates ATG codons to a stop codon (TGA) showed that only ATG number 3 (at position +251) was strictly required for the production of packaged GTA DNA, suggesting it represents the final *gtaT* start codon (Fig. 3b).

The identification of the TSS also allowed us to identify the likely −10 (TATCTTT) and −35 (AATCAT) of the GTA promoter (Fig. 3a). Using a *lacZ* reporter construct containing DNA from −153 to +229 relative to the *gtaT* TSS, we introduced mutations into the putative −10 and −35 elements to find that the β-galactosidase activities were abolished or reduced (Fig. 3c). Substituting the −35 region of P*gtaT* by a near consensus −35 region from highly expressed *rsaA* gene[45] caused ~sixfold reduction in β-galactosidase activities, however, this chimeric promoter was active regardless of the presence or absence of GafYZ (Supplementary Fig. 4). Overall, our mutational analysis established the importance of the −10 and −35 elements for the promoter activity of *gtaT*.

Next, we sought to define a consensus GafY binding motif and the position of the GafY binding site in relation to the IHF binding element (IBE1) within the GTA promoter. Due to the small number of enriched regions in the anti-GafY ChIP-seq dataset, it was difficult to predict a cognate GafY binding motif from MEME analysis. To overcome this, we sought to identify the minimal region of the GTA promoter that was responsive to GafY. To do so, we constructed a series of GTA promoter-*lacZ* fusions that contained progressively truncated DNA flanking the TSS (Fig. 4a). A DNA region ranging from −153 to +229 relative to the *gtaT* TSS, when fused to *lacZ*, produced a background level of β-galactosidase activity in the GTA-off WT background (Fig. 4a). In contrast, the same construct gave a high β-galactosidase activity in the GTA-on *ΔrogA* background (the activators GafYZ are produced in the absence of the RogA repressor, thus switching on GTA transcription) (Fig. 4a). This result suggested that the −153 to +229 region contains all the elements necessary for GafYZ-mediated

transcriptional activation. In the *ΔrogA* background, the β-galactosidase activities did not change drastically when 10, 20, or 30 bp were truncated from the upstream end of the P*gtaT* (−153 to +229)-*lacZ* fusion constructs (Fig. 4a). However, truncation of 40 bp reduced the β-galactosidase activity by ~20-fold (Fig. 4a), almost to the background level, suggesting a key regulatory element, possibly a GafY binding element, is in the DNA region from −123 to −113 relative to the *gtaT* TSS (Fig. 4a). Indeed, when mutations (P*gtaT* YBE*: ACTATG → TGACCC) were introduced into this putative GafY binding element (YBE), GafY binding was eliminated as assessed by anti-GafY ChIP-seq (Fig. 4b), providing further evidence that this represents the site of GafY binding.

## GafY interacts directly with the sigma factor RpoD

In the promoter of the GTA cluster, the IHF binding element is positioned between the core −10 −35 promoter and the far upstream GafY binding element, consistent with the classical role of IHF in bending DNA to facilitate the recruitment of RNA polymerase to the core promoter by an upstream transcriptional activator. We hypothesized that GafY might interact directly with a component of the RNA polymerase (RNAP) holoenzyme to activate transcription[46,47]. To identify potential interacting partners of GafY, we conducted an in silico AlphaFold2-Multimer-based protein interaction screen[48,49] between GafY and 23 core components of RNAP and sigma factors in *C. crescentus* (Fig. 5a). Based on confidence metrics (ipTM) generated by AlphaFold2, sigma factor 70 (RpoD), specifically its domains 3−4, were predicted to interact with GafY (Fig. 5a). To investigate this potential GafY-RpoD interaction, we performed a co-immunoprecipitation (co-IP) experiment using FLAG-tagged RpoD as bait. We observed that GafY was indeed pulled down with FLAG-RpoD, while another DNA-binding protein, ParB[50], serving as a negative control, was not (Fig. 5b). Furthermore, to assess whether GafY and RpoD form a direct complex, we co-overexpressed non-tagged GafY and His6-tagged *C. crescentus* RpoD in *Escherichia coli* (*E. coli*) (Fig. 5c). GafY was found to co-purify with His-tagged RpoD on a nickel affinity column (Fig. 5c). Similarly, GafY co-purified with domains 3−4 of RpoD alone, showing that GafY and RpoD interact directly via domains 3−4 of the sigma factor (Fig. 5c). Consistent with these observations, the promoter of the GTA cluster, as well as in the majority of the other GafY target promoters, were enriched in anti-FLAG-RpoD ChIP-seq experiments (Fig. 5d). Altogether, we suggest that GafY interacts with RpoD to recruit RNAP holoenzyme to the GTA promoter.

## Evidence that GafZ form a transcription elongation complex with RNAP, NusA, NusG, and NusE

We noted from Gozzi et al.[23] that the anti-GafZ ChIP-seq peak at the GTA promoter (its only target) is asymmetrical, extending downstream across the entire GTA cluster[23] (Fig. 6a). We hypothesized that GafZ associates with RNAP as it transcribes the GTA cluster. To further investigate this possibility, we performed anti-FLAG ChIP-seq experiments in the GTA-on *ΔrogA* strain harboring a FLAG-tagged β′ (RpoC)

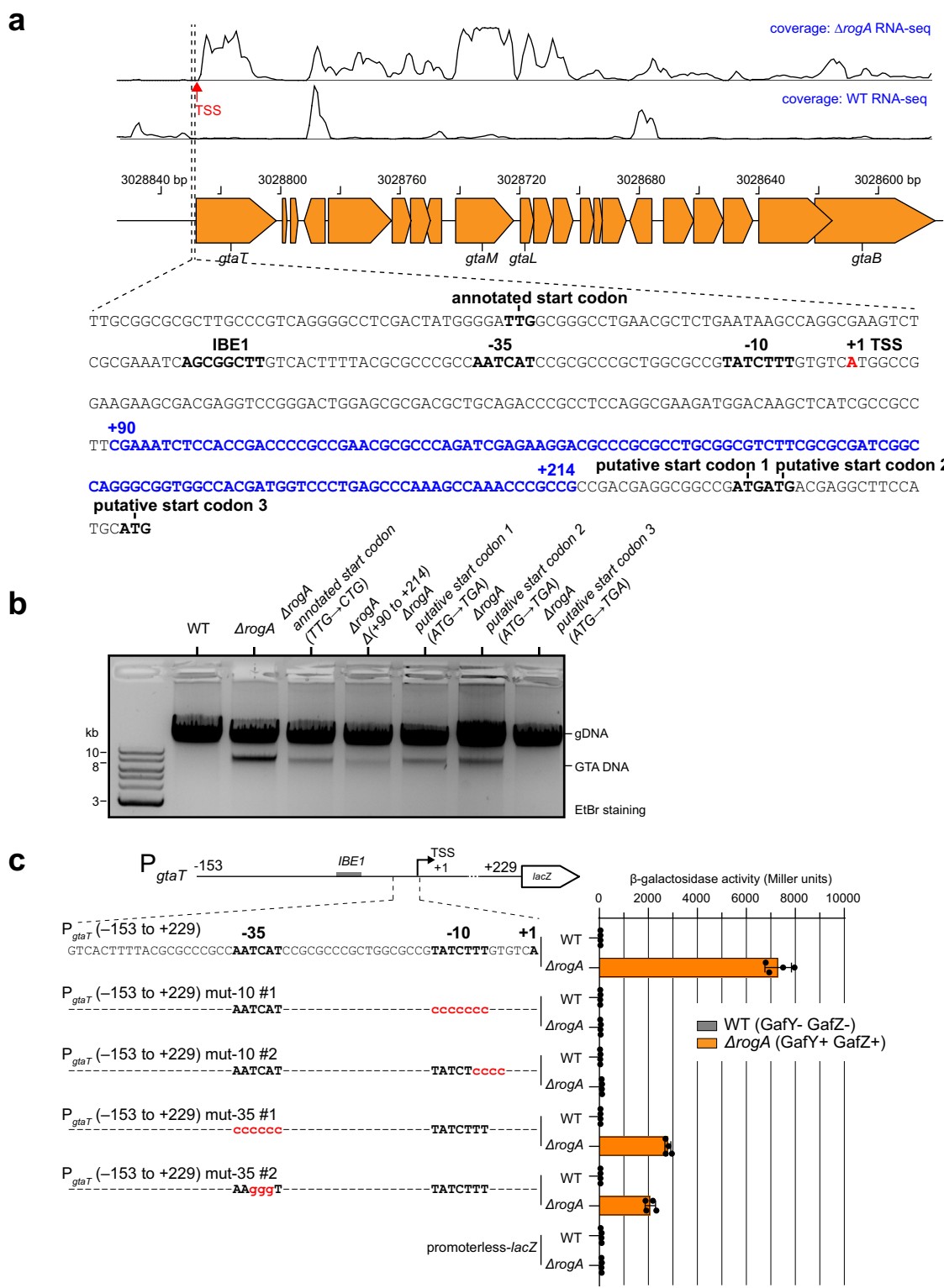

**Fig. 3 | Defining the transcription start site, core promoter elements, and the start codon for *gtaT*, the first gene in the GTA main cluster. a** Plots of RNA-seq data for stationary-phase GTA-on *ΔrogA* cells vs. GTA-off WT cells. A single prominent TSS (red arrow) was found for the GTA main cluster in the GTA-on cells. A sequence covering the upstream region of *gtaT* is shown beneath the schematic diagram of genes, and features such as the annotated/putative start codons, IHF-binding element (IBE), core promoter elements (−10 −35), and the transcription start site (TSS, red) are indicated. The sequence from +90 to +214 (blue) was deleted for the experiment reported in panel (**b**). **b** Total DNA extraction from

indicated strains grown up to a stationary phase. Experiments were performed at least twice, and a representative image is shown. **c** β-galactosidase activity (in Miller units) of the indicated promoter-*lacZ* reporter constructs in two *C. crescentus* genetic backgrounds (WT (gray) vs *ΔrogA* (orange)). The upstream region of *gtaT*, from −153 to +229 relative to the +1 TSS, was fused to *lacZ*. The −10 and −35 elements were mutated as indicated (red) on the schematic diagram. Cells containing an empty (promoterless) *lacZ* reporter plasmid served as a negative control. Values and error bars indicate mean ± standard deviations from four replicates. Source data are provided as a Source data file.

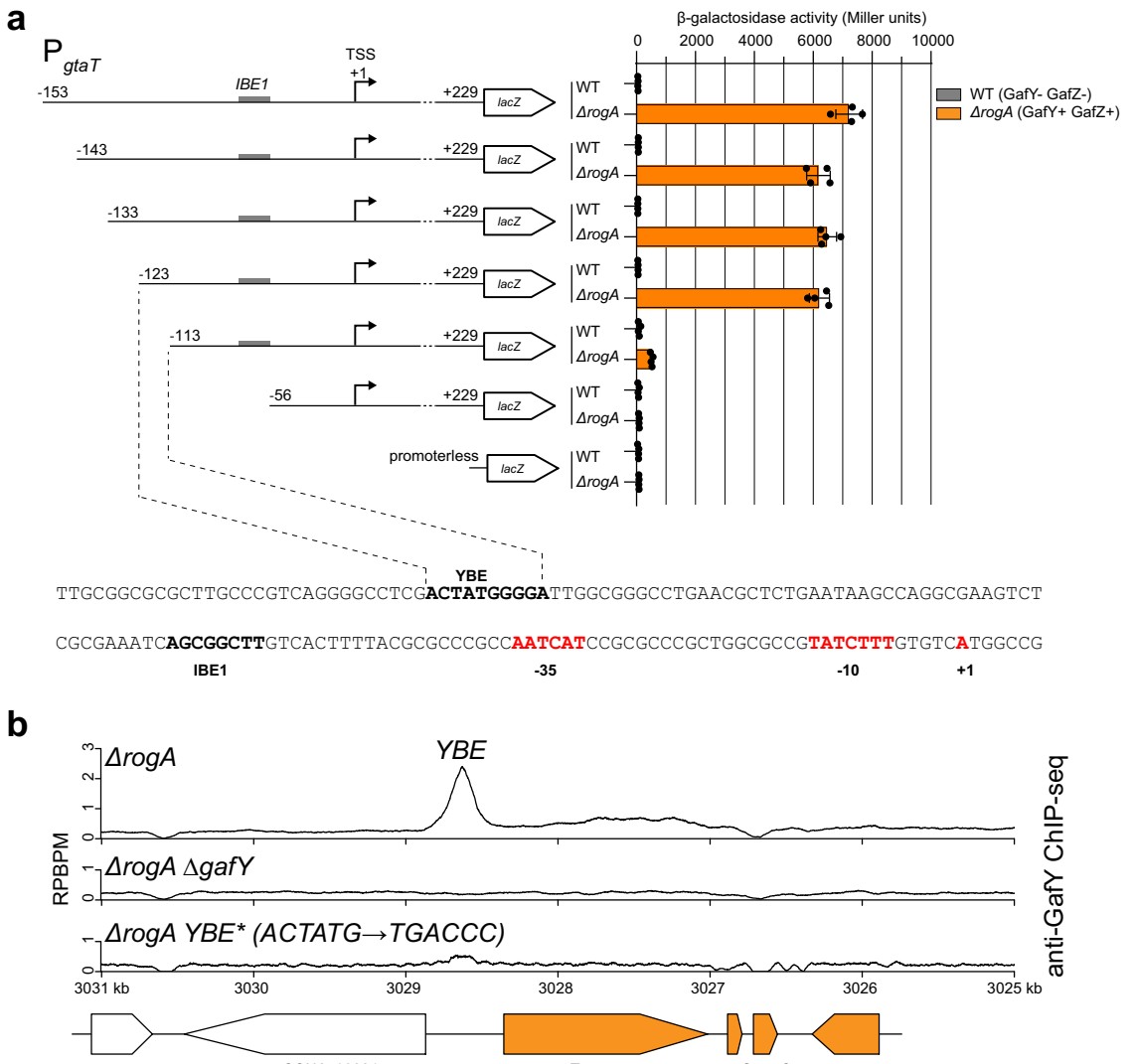

**Fig. 4 | A putative GafY-binding element (YBE) is located upstream of the IHF-binding element (IBE) on the promoter region of the main GTA cluster. a** β-galactosidase activity (in Miller units) of progressively truncated promoter-*lacZ* reporter constructs in two *C. crescentus* genetic backgrounds (WT (gray) vs *ΔrogA* (orange)). Values and error bars indicate mean ± standard deviations from four replicates. The sequence of a putative GafY-binding element (YBE) is shown together with the IHF-binding element (IBE1), the core promoter elements (−10 −35) (red), and the TSS. Source data are provided as a Source data file for (**a**). **b** Anti-GafY ChIP-seq profiles show the enrichment of GafY at the upstream region of the GTA main cluster in different genetic backgrounds. Profiles were plotted with the x-axis representing genomic positions and the y-axis representing the number of reads per base pair per million mapped reads (RPBPM). ChIP-seq experiments were performed twice using biological replicates, and a representative profile is shown. MACS2-identified anti-GafY ChIP-seq peak above YBE in *ΔrogA* sample is reproducible in both replicates and significant i.e., having Poisson distribution −log$_{10}$($p$ value) and false discovery rate −log$_{10}$($q$ value) > 900 in both replicates.

subunit of RNAP and compared the results to that from a negative control of a non-tagged *ΔrogA* strain (Fig. 6a). In parallel, we also performed anti-VSVG ChIP-seq experiments with *ΔrogA* cells expressing VSVG epitope-tagged NusA, NusG, or NusE, using the appropriate non-tagged negative controls (Fig. 6a). In all cases, the ChIP-seq profiles are highly similar and correlated, with signals spreading into and across the entire GTA gene cluster (Fig. 6a) (Pearson's correlation values of anti-FLAG-GafZ ChIP-seq profile (from 3010 kb to 3030 kb) vs. anti-RpoC-FLAG, anti-VSVG-NusAGE profiles are 0.64, 0.93, 0.95, and 0.92, respectively, $p < 10^{-16}$). These data suggest that GafZ, together with RNAP and Nus proteins, might form an elongation complex that transcribes the main GTA gene cluster. To investigate this possibility further, we engineered *ΔrogA* FLAG-*gafZ* strains that harbor individually VSVG-tagged NusA, NusG, NusE, and performed co-IP using FLAG-tagged GafZ as bait (Fig. 6b). We observed an enrichment of Nus proteins in the IP fraction compared to the pre-IP input control (Fig. 6b). Another DNA-binding protein, ParB, was not pulled down in

the IP fraction (Fig. 6b), suggesting that the enrichment of Nus proteins was specific.

**The putative GafZ binding element (ZBE) is located in between the −10 and −35 promoter elements of the GTA gene cluster**

Despite the previous observation that GafY and GafZ interact with each other[23], unlike GafY, GafZ has only one target—the promoter of the GTA cluster[23] (Fig. 6a). This suggested there must be a GafZ binding element that recruits GafZ to the promoter of the GTA cluster. Further, AlphaFold2 and FoldSeek[48,51] predict that GafZ contains a sigma-factor-like helix-turn-helix motif, consistent with direct DNA binding (Supplementary Fig. 5). We inspected the sequence of the GTA promoter for inverted repeats that might represent a GafZ binding element (ZBE), and identified a near-perfect palindrome (GCGCCCG-CTGGCGC) in between the −10 and −35 elements (Fig. 7a). Next, we introduced mutations (P$_{gtaT}$ ZBE*: CTGGCGC → TTTTTTC) to the right half of this palindrome and found

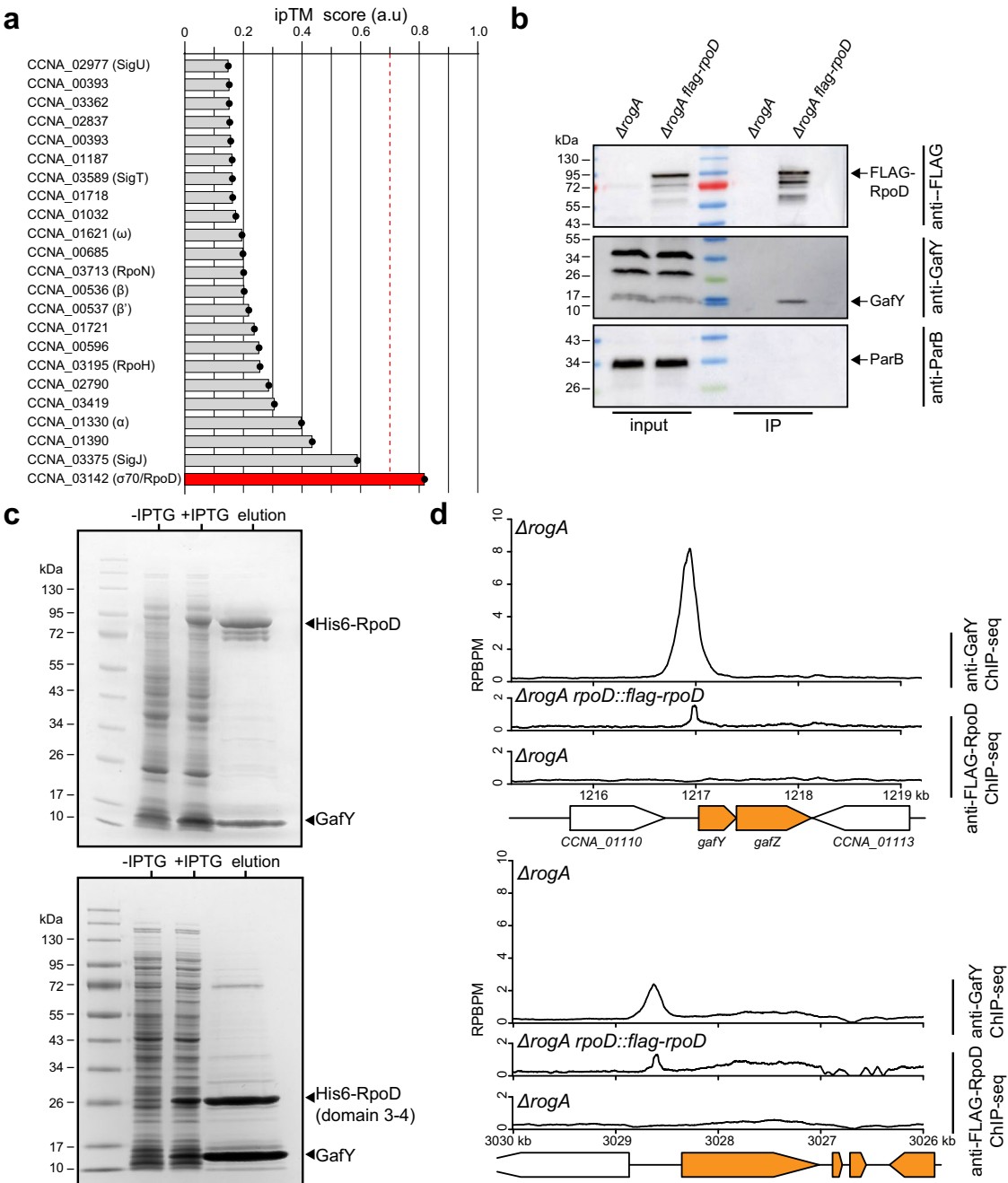

**Fig. 5 | GafY interacts directly with the sigma factor 70 RpoD. a** An AlphaFold2-Multimer-based protein interaction screen between GafY and 23 core components of *C. crescentus* RNAP and sigma factors suggests a GafY-RpoD interaction. The likelihood of protein−protein interaction was ranked based on the confidence metrics (ipTM), with ipTM greater than 0.7 indicating likely protein−protein interaction. **b** Immunoblots analysis of co-immunoprecipitation of FLAG-tagged RpoD. ParB served as a non-associated protein control. The positions of bands corresponding to FLAG-tagged RpoD, GafY, and ParB are indicated with arrows. Co-IP experiments were performed at least twice, and a representative image is shown. **c** Co-expression and purification of a soluble RpoD-GafY complex (top panel) or RpoD (domain3−4 only)-GafY (bottom panel). RpoD (full-length or truncated) and GafY were co-overexpressed in *E. coli*, with only RpoD or RpoD (domain 3−4) carrying an N-terminal His tag. Following co-overexpression, the soluble extract was passed over a nickel column, and, after washing, bound proteins were eluted and analyzed on SDS-PAGE. Experiments were performed at least twice, and a representative image is shown. **d** Anti-GafY and anti-FLAG ChIP-seq profiles show the enrichment of GafY and FLAG-tagged RpoD, respectively, at the upstream region of *gafYZ* (top panel) and the GTA main cluster (bottom panel) in different genetic backgrounds. Profiles were plotted with the x-axis representing genomic positions and the y-axis representing the number of reads per base pair per million mapped reads (RPBPM). ChIP-seq experiments were performed twice using biological replicates, and a representative profile is shown. MACS2-identified anti-GafY and anti-FLAG-RpoD ChIP-seq peaks in this panel are reproducible in both replicates and significant i.e., having Poisson distribution $-\log_{10}(p$ value) and false discovery rate $-\log_{10}(q$ value) >1000 and >140, respectively, in both replicates. Source data are provided as a Source data file.

that the enrichment of FLAG-GafZ was reduced to background level (Fig. 7a). In contrast, the enrichment of GafY in the upstream promoter region was largely unaffected (Fig. 7b). However, GafY was no longer enriched in the coding region of *gtaT* and the main GTA cluster (Fig. 7b), possibly because GafZ and the GafZ-GafY complex failed to form a transcription elongation complex with RNAP when ZBE was mutated. Lastly, the $P_{gtaT}$ ZBE* mutations also eliminated the production of GTA-packaged DNA and GtaL protein (Fig. 7c).

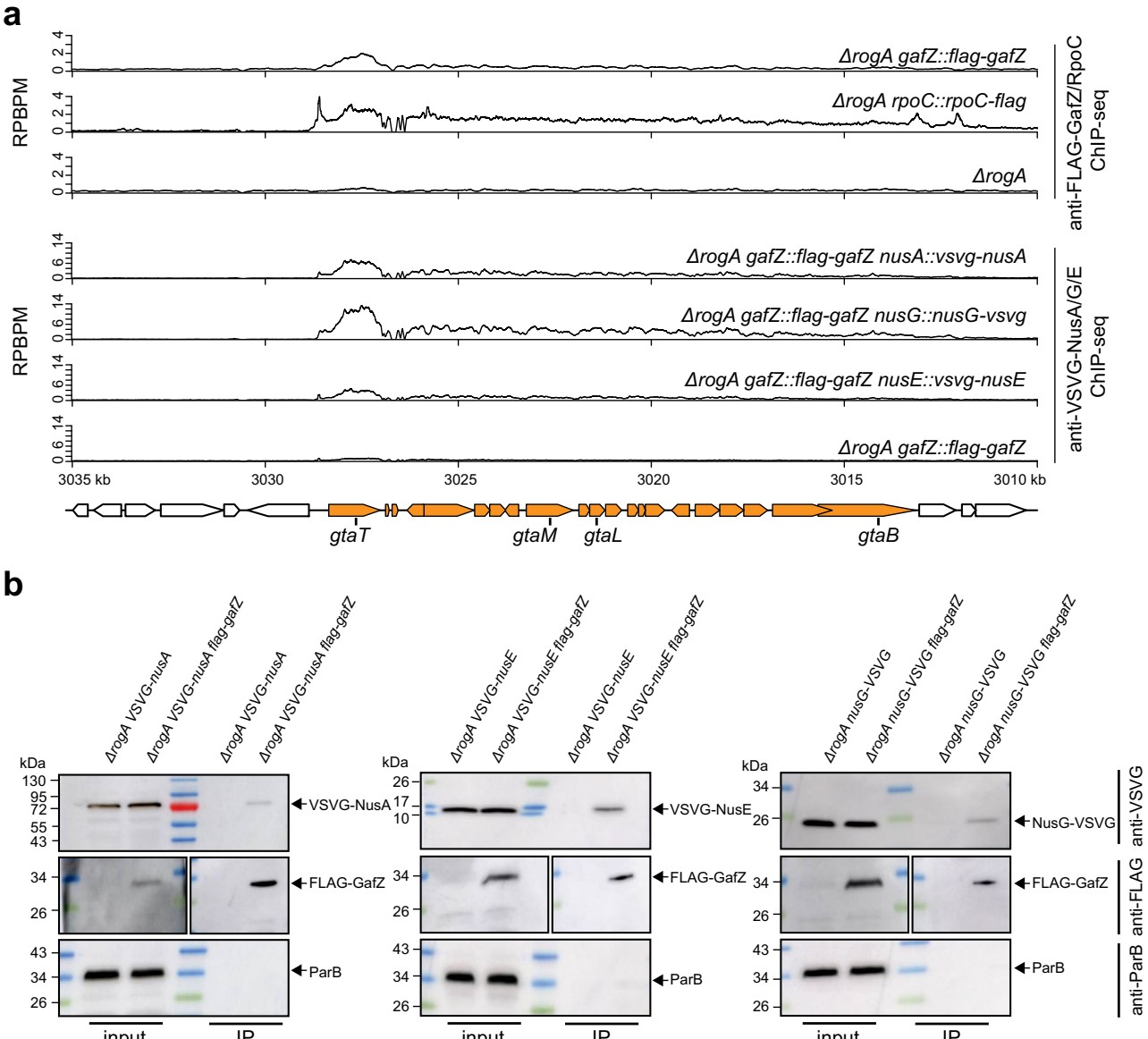

**Fig. 6 | GafY and GafZ might form a transcription elongation complex with RNAP, NusA, NusG, and NusE. a** Anti-FLAG and anti-VSVG ChIP-seq profiles show the enrichment of FLAG-tagged GafZ and VSVG-tagged NusAGE, respectively, at the GTA main cluster in different genetic backgrounds. Profiles were plotted with the x-axis representing genomic positions and the y-axis representing the number of reads per base pair per million mapped reads (RPBPM). ChIP-seq experiments were performed twice using biological replicates, and a representative profile is shown. MACS2-identified anti-FLAG-RpoC and anti-VSVG-NusAGE ChIP-seq peaks in this panel are reproducible in both replicates and significant i.e., having Poisson distribution −log$_{10}$($p$ value) and false discovery rate −log$_{10}$($q$ value) >800 and >290, respectively, in both replicates. **b** Immunoblots analysis of co-immunoprecipitation of FLAG-tagged GafZ. ParB served as a non-associated protein control. The positions of bands corresponding to VSVG-tagged NusA/E/G, FLAG-tagged GafZ, and ParB are indicated with arrows. The immunoblot for anti-FLAG of the input fraction was exposed for a longer time than that from the IP fraction as FLAG-tagged GafZ was highly concentrated in the IP fraction compared to the input fraction. Co-IP experiments were performed at least twice, and a representative image is shown. Source data are provided as a Source data file.

Taken together, these findings demonstrated that the inverted repeat positioned between the −10 and −35 elements of the *gtaT* promoter is critical for GafZ binding and GTA synthesis.

**GafYZ allows RNAP to bypass a transcription terminator located downstream of the first gene in the GTA gene cluster**
We noted a sharp reduction in signal in ChIP-seq datasets of anti-FLAG-GafZ, anti-GafY, anti-FLAG-β' RNAP, and anti-VSVG-NusAEG immediately downstream of *gtaT*, the first gene of the GTA cluster (Fig. 6a and Fig. 8a). Focusing on this region, we discovered three 150-bp long GC-rich direct repeats, each of which contains imperfect inverted repeats. We hypothesized that these repeats might

form a putative transcription terminator (*ter$_{GTA}$*) (Fig. 8a and Supplementary Fig. 6a). Given the association of GafYZ with the RNAP elongation complex and the strict requirement of GafZ for transcription of the GTA cluster, we reasoned that GafYZ might enable RNAP to read through this putative *ter$_{GTA}$* terminator. To investigate these possibilities, we determined the effect of this putative *ter$_{GTA}$* on transcription activities of a strong GafYZ-independent promoter (P$_{rsaA}$, driving the expression of the most abundant S-layer protein-encoding gene, *rsaA*, in *C. crescentus*[45,52]) and of a GafYZ-dependent *gtaT* promoter, using promoter-*lacZ* fusion reporters (Fig. 8b). We fused a DNA region ranging from −153 to +229 relative to the *gtaT* TSS to *lacZ*. To assay for β-galactosidase

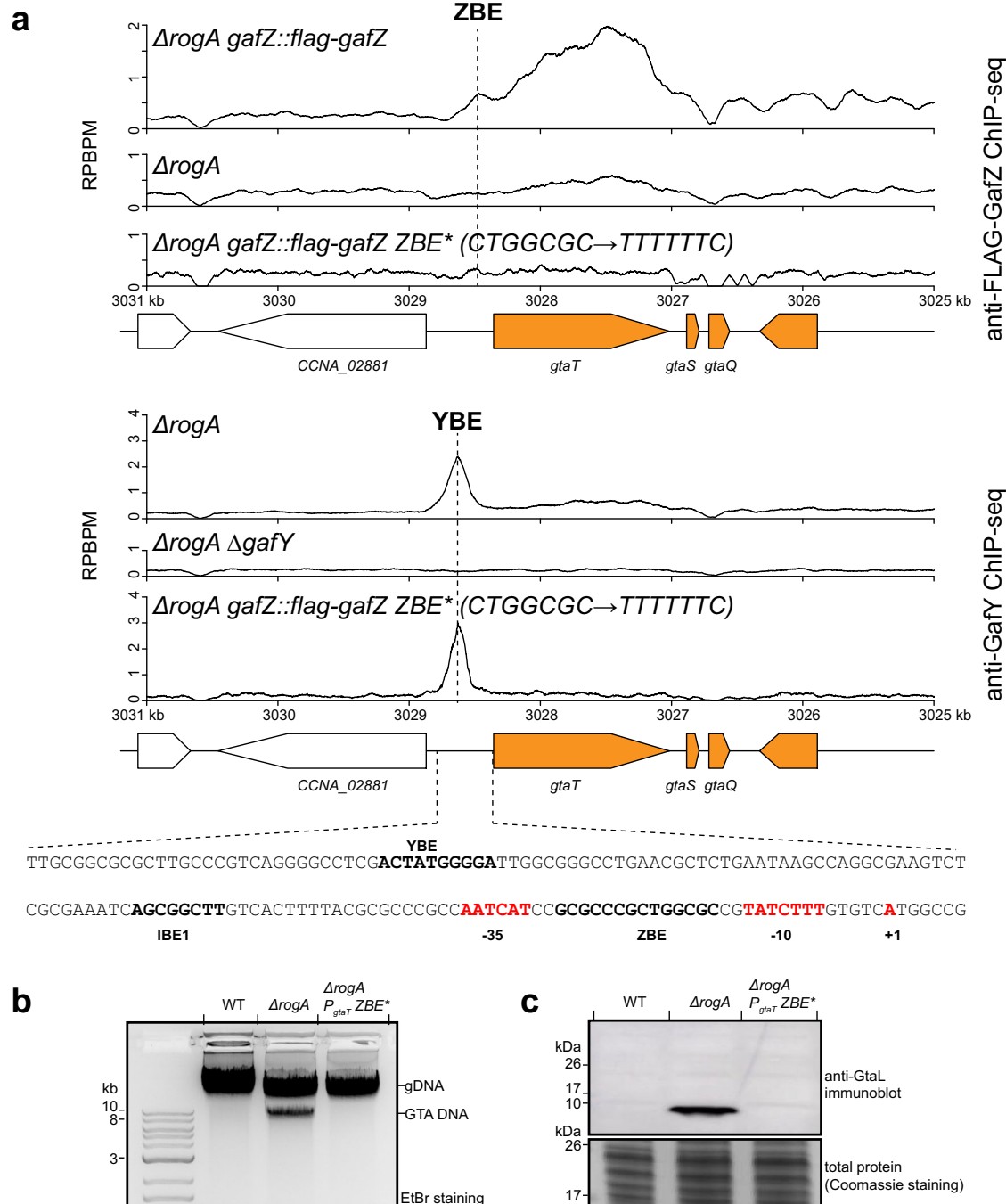

**Fig. 7 | The putative GafZ binding element (ZBE) is located in between the −10 and −35 promoter elements of the GTA gene cluster. a** Anti-FLAG and anti-GafY ChIP-seq profiles show the enrichment of FLAG-tagged GafZ and GafY, respectively, at the main GTA cluster in different genetic backgrounds. Profiles were plotted with the x-axis representing genomic positions and the y-axis representing the number of reads per base pair per million mapped reads (RPBPM). The sequence of a putative GafZ-binding element (ZBE) is shown together with the IHF-binding element (IBE1), the GafY-binding element (YBE), the core promoter elements (−10 −35), and the TSS. ChIP-seq experiments were performed twice using biological replicates, and a representative profile is shown. MACS2-identified anti-FLAG-GafZ and anti-GafY ChIP-seq peaks in this panel are reproducible in both replicates and significant i.e., having Poisson distribution $-\log_{10}(p$ value) and false discovery rate $-\log_{10}(q$ value) > 400 in both replicates. **b** Total DNA extraction from indicated strains grown up to a stationary phase. Total DNA was purified and separated by electrophoresis on a 1% agarose gel which was stained with ethidium bromide (EtBr) for DNA. Experiments were performed at least twice, and a representative image is shown. Source data are provided as a Source data file for (**b**). **c** Immuno**b**lot of total cell lysates of indicated strains using a polyclonal anti-GtaL (GTA head-tail connector protein) antibody. A separate Coomassie-stained SDS-PAGE was loaded with the same volume of samples to serve as a loading control. Experiments were performed at least twice, and a representative image is shown.

activity, reporter plasmids were introduced into three genetic backgrounds, namely a WT GTA-off strain (GafY- GafZ-), a *ΔrogA* GTA-on strain (GafY+ GafZ+), and a *ΔrogAΔgafZ* GTA-off strain (GafY+ GafZ-) (Fig. 8b).

A P*rsaA-lacZ* fusion construct produced high levels of β-galactosidase activity in all three genetic backgrounds (Fig. 8b), consistent with P*rsaA* being independent of GafYZ (no GafY or GafZ binding sites were found in the upstream region of *rsaA* by ChIP-seq). Insertion

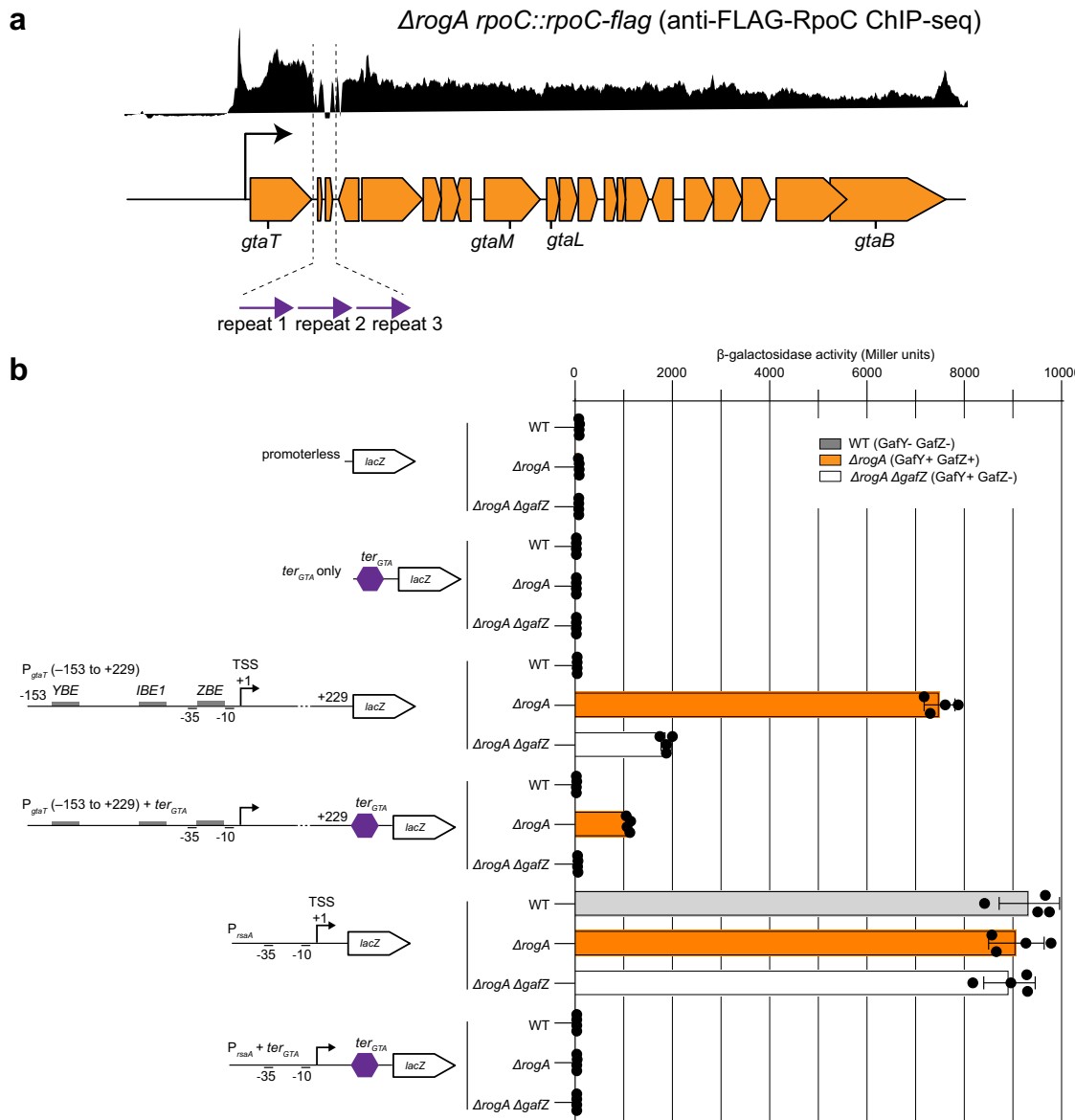

**Fig. 8 | GafYZ allows RNAP to bypass a transcription terminator located downstream of the first gene in the GTA gene cluster. a** Identification of three 150-bp direct repeats (purple arrows) downstream of the *gtaT* gene in the GTA main cluster (see Supplementary Fig. 6a for the sequences and alignment of the three repeats). **b** β-galactosidase activity (in Miller units) of indicated promoter-*lacZ* reporter constructs in three *C. crescentus* genetic backgrounds (WT (gray) vs *ΔrogA* (orange) vs. *ΔrogAΔgafZ* (white)). The upstream region of *gtaT*, from −153 to +229 relative to the +1 TSS, was fused to *lacZ*, with or without the three direct repeats (*ter_GTA*, purple hexagon). Cells containing an empty (promoterless) *lacZ* reporter plasmid or plasmid with only *ter_GTA* fused to *lacZ* served as negative controls. Values and error bars indicate mean ± standard deviations from four replicates. Source data are provided as a Source data file.

of the three 150-bp repeats in between P_*rsaA* and *lacZ* reduced β-galactosidase activity to background levels regardless of the presence of GafYZ (Fig. 8b), confirming that the tandem repeats constitute a strong transcription terminator.

A P_*gtaT* (−153 + 229)-*lacZ* fusion was active in a GTA-on *ΔrogA* (GafY + GafZ+) background but was reduced to the background level when both GafY and GafZ were absent, and ~4-fold reduced when only GafY was present i.e., in the *ΔrogAΔgafZ* GTA-off (GafY+ GafZ-) background (Fig. 8b). An insertion of the putative *ter_GTA* in between P_*gtaT* (−153 + 229) and *lacZ* in the GTA-on *ΔrogA* (GafY+ GafZ+) background reduced the β-galactosidase activity by ~8-fold but, crucially, did not eliminate all transcriptional activity. On the other hand, only background level β-galactosidase activity was detected for the same construct when GafZ was absent (Fig. 8b). We further observed a similar anti-termination response by GafZ when we replaced the putative

*ter_GTA* by a well-characterized T1 transcription terminator from the *E. coli rrnB*[53] (Supplementary Fig. 6b). Overall, our results demonstrated that the tandem repeats function as a transcription terminator, and that GafZ acts as a transcription anti-terminator, allows some (but not all) RNAP to read through and transcribe the entire GTA cluster.

## Discussion
In this study, we demonstrate that the *C. crescentus* GTA gene cluster is transcriptionally co-activated by IHF and GafY, and by a GafYZ-mediated transcription anti-termination. We show that, in the absence of the repressor RogA, IHF and GafY directly bind to the promoter region of the *gafYZ* operon, the promoter of the main CcGTA cluster, and ~18 promoters of accessory GTA genes, to activate transcription (Fig. 9a). Our findings support a model that GafY interacts directly with domain 3–4 of the housekeeping sigma factor 70 (RpoD) to assist

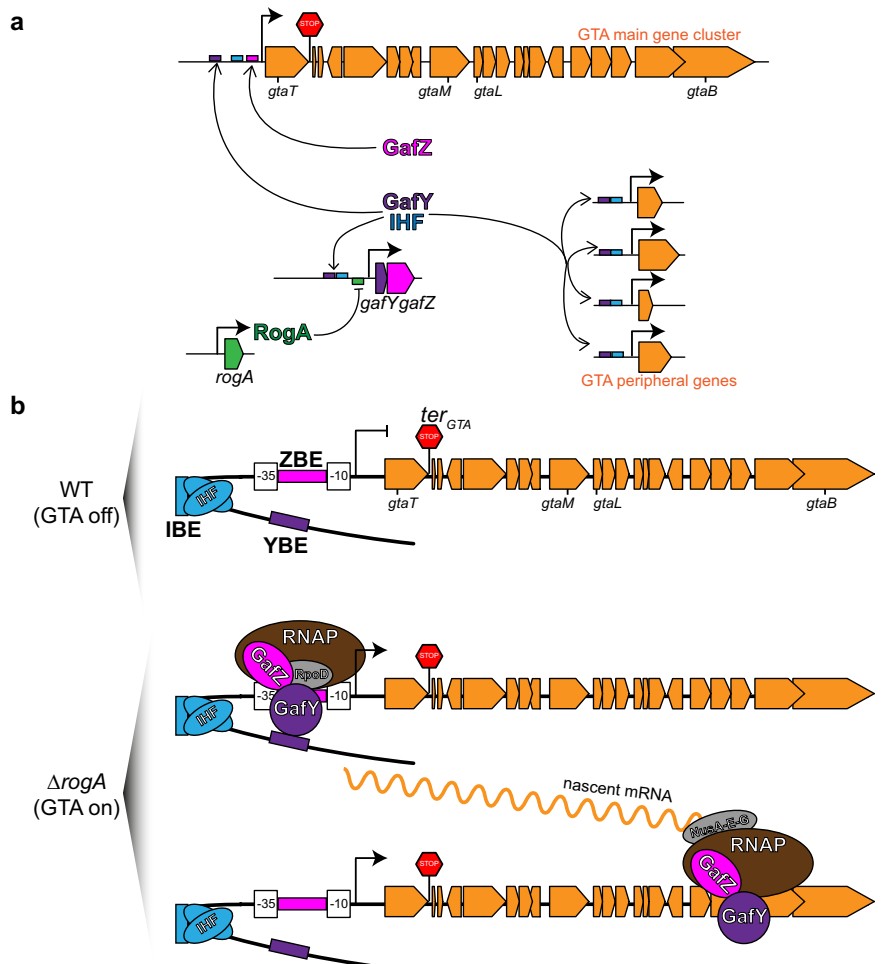

**Fig. 9 | Proposed model of how GafYZ and IHF function to activate the GTA gene cluster. a** Diagram of CcGTA regulation. RogA (dark green) directly represses (flat-headed arrow) the expression of *gafYZ*. GafY (purple) and IHF (blue) co-activate (arrows) the *gafYZ* operon, the GTA main gene cluster (orange) as well as GTA peripheral genes (orange) on the *C. crescentus* chromosome. GafZ (magenta) together with GafY and IHF are required to activate the transcription of the GTA main cluster. **b** In GTA-off WT cells, RogA represses the expression of *gafYZ*. No GafY and GafZ were produced to activate the transcription of the GTA main gene cluster, even though IHF (blue) is prebound at the upstream region of the GTA main cluster at its IHF-binding element (IBE). In GTA-on Δ*rogA* cells, the promoter of

*gafYZ* is derepressed, leading to the production of GafY (purple) and GafZ (magenta). GafY binding at the putative GafY-binding element (YBE), likely being assisted by the DNA-bending action of IHF (blue), binds RpoD (gray) to recruit RNAP (brown) to the core −10 −35 promoter elements. GafZ (magenta) binds the promoter at the putative GafZ-binding element (ZBE) positioned between the −10 and −35 elements of the promoter, then binds RNAP to allow a holo-complex of RNAP-GafZ-NusAEG to bypass the *ter*$_{GTA}$ (red stop sign) positioned downstream of *gtaT* to synthesize nascent mRNA (wavy orange line). ChIP-seq profiles of GafY and biochemical data that show GafY-GafZ co-interaction suggest that GafY might travel together with an elongating RNAP-GafZ-NusAEG holo-enzyme complex.

RNAP binding to a degenerate −35 element at these promoters. Supporting this model, substituting the −35 region of P$_{gtaT}$ by a consensus −35 region from a sigma factor 70-dependent promoter (P$_{rsaA}$) circumvented the need for a transcription activator, resulting in constitutive expression (Supplementary Fig. S4). Most transcription activators that interact with domain 3–4 of sigma factor 70 bind DNA close to the −35 element of their target promoters[54], however, we noted that GafY binds far upstream of the −35 region and strictly requires IHF for transcription activation (Fig. 9b). IHF is a known DNA-binding and bending transcriptional factor[39–42], and it has been well-established that IHF-induced DNA bending enables bacterial enhancer-binding proteins (bEBPs), which bind far upstream of the core promoter region, to contact RNAP-sigma factor 54 holo-enzyme pre-recruited on the promoter to activate transcription[54,55]. We reason that IHF might perform a similar DNA-bending role to enable GafY to loop over to contact domain 3–4 of sigma factor 70 (Fig. 9b). Given our evidence that GTA-related promoters are sigma factor 70-dependent (not sigma factor 54) (Fig. 5), it is rare that IHF is required for transcription activation by a co-activator that binds far upstream of the −35

element of a sigma factor 70-dependent promoter. Other than the case reported in this work, the only other documented example is the activation of an *E. coli* nitrate reductase (*narGHJI*) operon that requires NarL, FNR, and IHF[56]. Lastly, we hypothesize that the spacing between the GafY-binding element (YBE) and the IHF-biding element (IBE) is crucial for DNA looping and the subsequent transcription activation of GTA promoters, future experiments that change the helical position of YBE and the YBE-IBE distance are necessary to test this hypothesis.

In this work, we also present evidence that GafZ is a transcription anti-terminator that likely forms an anti-termination complex with RNA polymerase, NusA, NusG, and NusE to bypass transcription terminators downstream of the first gene in the CcGTA cluster, resulting in transcription of the entire CcGTA cluster gene (Fig. 9b). It is also worth noting that deleting *gafZ* from Δ*rogA* cells reduced the β-galactosidase activity of the P$_{gtaT}$ (−153 to +229)-*lacZ* reporter by ~fourfold, even though GafY and IHF were present and this reporter construct does not contain *ter*$_{GTA}$ (Fig. 8b). It is possible that GafZ might also increase the processivity of RNAP to elongate through a 229-nt long untranslated region, in addition to bypassing transcription

terminators later. Here, we also identified a putative DNA element (ZBE) important for the regulation of the main CcGTA cluster by GafZ. This putative ZBE is located between the −10 and −35 elements of the CcGTA cluster promoter, similar to the location of the binding elements for well-characterized processive anti-terminators such as Q from bacteriophage λ and AlpA from *Pseudomonas aeruginosa*[57,58]. The ChIP-seq profiles of *C. crescentus* GafZ are also reminiscent of the binding patterns of RNAP and transcription anti-terminators such as *P. aeruginosa* AlpA, *E. coli* RfaH, and λQ[57,59–61]. The λQ binding element (QBE) and the AlpA binding element (ABE) help the direct loading of Q and AlpA onto RNAP[57,58,62–65], and it seems likely that the putative GafZ binding element (ZBE) in the GTA promoter may help loading of GafZ onto RNAP in a similar way, allowing RNAP to bypass the long untranslated region and downstream transcriptional terminators (Fig. 9b). The cryo-EM structures of λQ- and AlpA-bound RNAP have been solved, showing that these anti-terminators form a molecular nozzle near the RNA-exit channel of RNAP to prevent the formation of terminator hairpin structures that would otherwise form in the nascent RNA and thereby impede or stop transcription elongation[58,62,66]. While AlphaFold-predicted structures of GafZ show no sequence or structural similarity to these known anti-terminators, it has been shown recently that Q protein of bacteriophage 21 (Q21) and λQ, despite sharing no sequence similarity, both form a nozzle that narrows and extends the RNAP RNA-exit channel to prevent the formation of RNA hairpin[62]. Future works, especially solving a cryo-EM structure of GafYZ-RNAP-DNA holo-enzyme complex, will hope to determine how GafZ modifies RNAP to mediate transcription anti-termination and whether GafZ shares the same mechanism as AlpA, Q21, and λQ proteins.

We observed that GafY and GafZ have different promoter specificity, GafZ has only one target—the promoter of the main GTA cluster, while GafY regulates ~18 GTA-related promoters (Fig. 9a). This is seemingly at odds with the previous observation that GafYZ form a complex[23]. However, it is not yet known how stable or transient GafYZ complex is. Furthermore, while both GafY and GafZ have predicted DNA-binding domains of their own, only the promoter of the main GTA cluster has a dedicated GafZ-binding element (ZBE). We speculate that this additional GafZ-ZBE DNA interaction contributes to the selection of promoter.

GafY and GafZ show sequence homology to the N-terminal and C-terminal domain, respectively, of *R. capsulatus* GafA, the direct activator of RcGTA[36]. Although the exact binding elements for *R. capsulatus* GafA have not been mapped at nucleotide resolution, Sherlock and Fogg (2022) showed using EMSA assays that C-terminal domain of GafA (equivalent to *C. crescentus* GafZ) binds to a DNA fragment covering the −10, −35 and TSS of the RcGTA cluster[37]. Given that *R. capsulatus* GafA and *C. crescentus* GafZ are homologous, and the similar location of their DNA-binding elements, it is possible that *R. capsulatus* GafA might have previously unrecognized transcription anti-termination activity. Sherlock and Fogg (2022) demonstrated that the central region in between the N-terminal and C-terminal domains of GafA interacts with the omega (RpoZ-ω) subunit of RNAP, thereby recruiting RNAP to activate the RcGTA cluster[37]. In *C. crescentus*, however, we have not observed the enrichment of VSVG-tagged RpoZ-ω in a co-IP using FLAG-tagged GafZ as bait (Supplementary Fig. 7). A sequence alignment of GafA and GafY-Z shows that the central region of GafA has the least similarity to a fusion of GafY and GafZ (Supplementary Fig. 8). Furthermore, several predicted loops in the central region of GafA are missing in the *C. crescentus* GafYZ fusion (Supplementary Fig. 8). This likely explains why *C. crescentus* GafYZ do not appear to interact with RpoZ-ω. In *R. capsulatus*, (p)ppGpp, which is likely to interact directly with RNAP via RpoZ-ω, contributes to the synthesis of GTA particles[37,67]. For example, deletion of *rpoZ* or *relA/spoT* (responsible for the synthesis of (p)ppGpp) in *R. capsulatus*

reduced GTA synthesis by five-fold[67], while deletion of the sole *relA/spoT* homolog in *C. crescentus*[68,69] only reduces CcGTA production two-fold (Supplementary Fig. 7). Future work is necessary to better understand the possible role of ppGpp(p) and/or RpoZ-ω in CcGTA production in *C. crescentus*.

The discovery that *C. crescentus* produces bona fide GTA particles offered a new and highly tractable organism to dissect the function, biosynthesis, and regulation of these enigmatic genetic elements[23]. Here, we revealed that GTA cluster gene expression is controlled by both transcriptional activation and by anti-termination. While most of the control, as revealed in this study, is at the cluster-specific level, it is interesting to note that a globally acting factor, IHF, is co-opted in the activation of the CcGTA gene cluster. In future work, we hope to gain further insight into the extent to which GTA is domesticated and integrated with the host's physiology, potentially shedding light on the evolution of such domestication. Lastly, *C. crescentus* GTA is not produced under normal laboratory conditions[23], and so it was necessary for us to exploit a *ΔrogA* strain to induce CcGTA production in this study. Finding the environmental or physiological signals that naturally de-repress RogA or activate GafYZ expression, if they exist, will be illuminating in understanding the benefit of GTA to the host and how such exaptation can evolve.

## Methods

### Strains, media, and growth conditions
*E. coli* and *C. crescentus* were grown in LB and PYE, respectively. When appropriate, media were supplemented with antibiotics at the following concentrations (liquid/solid media for *C. crescentus*; liquid/solid media for *E. coli* [μg/mL]): kanamycin (5/25; 30/50); spectinomycin (25/100; 50/50); oxytetracycline (1/2; 12/12). All strains, plasmids, and oligonucleotides used in this study are listed in Supplementary Data 2. Details on constructions of plasmids and strains are in the Supplementary Information. All plasmids and strains generated in this study are available upon request.

### Transposon (Tn5) mutagenesis
The Tn5 transposon delivery plasmid (pMCS1-Tn5-ME-R6Kγ-kanamycin^R-ME)[70] was conjugated from an *E. coli* S17-1 donor into *C. crescentus ΔrogA ΔlacA gtaM::gtaM-lacA* cells. Briefly, *E. coli* S17-1 was transformed with the transposon delivery plasmid and plated out on LB plates supplemented with kanamycin. On the next day, colonies forming on LB + kanamycin were scraped off the plates and resuspended in PYE to OD$_{600}$ of 1.0. Cells were pelleted down and resuspended in fresh PYE twice to wash off residual antibiotics. 100 μl of cells were mixed with 1000 μl of exponentially growing *C.crescentus ΔrogA ΔlacA gtaM::gtaM-lacA* cells then the mixture was centrifuged at 17,000 × *g* for 1 min. The cell pellet was subsequently resuspended in 50 μl of fresh PYE and spotted on a nitrocellulose membrane resting on a fresh PYE plates. PYE plates with nitrocellulose disks were incubated at 30 °C for 5 h before being resuspended by vortexing vigorously in fresh PYE liquid to release bacteria. Resuspended cells were plated out on Petri disks containing PYE agar supplemented with kanamycin and carbenicillin and 40 μg/ml X-gal, and incubated for 3 days at 30 °C. After 3-day incubation, white colonies were picked and restruck on PYE + kanamycin + carbenicillin + X-gal to purify. To locate the Tn5 insertion point, genomic DNA from mutants of interest was extracted. 4 μg of extracted genomic DNA was partial digested with Sau3AI in 50 μL reaction and re-ligated with T4 DNA ligase. The ligation mixture were ethanol precipitated and introduced into *E. coli pir116* cells by electroporation. Colonies carrying the re-ligated Tn5 plasmid grew on kanamycin and the plasmid was subsequently extracted and sequenced with oligo KAN-2 FP-1: ACCTACAACAAAGCTCTCATCAACC and R6KAN-2 RP-1:CTACCCTGTGGAACACCTACATCT to map the position of Tn5 insertion on the *C. crescentus* chromosome.

## Chromatin immunoprecipitation with deep sequencing (ChIP-seq)

*C. crescentus* cell cultures (50 mL) were grown in PYE to a stationary phase and fixed with formaldehyde to a final concentration of 1%. Fixed cells were incubated at room temperature for 30 min, then quenched with 0.125 M glycine for 15 min. Cells were washed three times with 1x PBS (pH 7.4) and resuspended in 1 mL of buffer 1 (20 mM K-HEPES pH 7.9, 50 mM KCl, 10% glycerol, and Roche EDTA-free protease inhibitors). Subsequently, the cell suspension was sonicated on ice using a Soniprep 150 probe-type sonicator (11 cycles, 15 s ON, 15 s OFF, at setting 8) to shear the chromatin to below 1 kb, and the cell debris was cleared by centrifugation (20 min at $17,000 \times g$ at 4 °C). The supernatant was then transferred to a new 2 mL tube and the buffer conditions were adjusted to 10 mM Tris-HCl pH 8, 150 mM NaCl and 0.1% NP-40. Fifty microliters of the supernatant were transferred to a separate tube for control (the input fraction) and stored at −20 °C. In the meantime, antibodies-coupled beads were washed off storage buffers before being added to the above supernatant. We employed anti-VSV-G antibody coupled to sepharose beads (Merck) for ChIP-seq of NusG-VSVG, VSVG-NusA, VSVG-NusE, and anti-FLAG antibody coupled to agarose beads (Merck) for ChIP-seq of RpoC-FLAG and FLAG-GafZ.

Briefly, 50 μL of anti-VSVG beads or 100 μL anti-FLAG beads was washed off storage buffer by repeated centrifugation and resuspension in IPP150 buffer (10 mM Tris-HCl pH 8, 150 mM NaCl and 0.1% NP-40). Beads were then introduced to the cleared supernatant and incubated with gentle shaking at 4 °C overnight. For anti-GafY ChIP-seq experiments, protein A sepharose beads (Merck) were incubated with the cleared supernatant for an hour to remove non-specific binding. Afterward, the cleared supernatant was retrieved and incubated with 50 μL of anti-GafY polyclonal antibody overnight. On the next day, protein A sepharose beads were added and incubated for 4 h to capture GafY-DNA complexes. Beads were then washed five times at 4 °C for 2 min each with 1 mL of IPP150 buffer, then twice at 4 °C for 2 min each in 1x TE buffer (10 mM Tris-HCl pH 8 and 1 mM EDTA). Protein-DNA complexes were then eluted twice from the beads by incubating the beads first with 150 μL of the elution buffer (50 mM Tris-HCl pH 8, 10 mM EDTA, and 1% SDS) at 65 °C for 15 min, then with 100 μL of 1X TE buffer + 1% SDS for another 15 min at 65 °C. The supernatant (the ChIP fraction) was then separated from the beads and further incubated at 65 °C overnight to completely reverse crosslinks. The input fraction was also de-crosslinked by incubation with 200 μL of 1X TE buffer + 1% SDS at 65 °C overnight. DNA from the ChIP and input fraction were then purified using a PCR purification kit (Qiagen) according to the manufacturer's instruction, and then eluted out in 40 μL water. Purified DNA was then constructed into libraries suitable for Illumina sequencing using the NEXT Ultra II library preparation kit (NEB). ChIP libraries were sequenced on the Illumina Hiseq 2500 or Nextseq 550 at the Tufts University Genomics facility.

## Processing ChIP-seq data

For analysis of ChIP-seq data, Hiseq 2500 or NextSeq 550 Illumina short reads (50 bp/75 bp) were mapped back to the *C. crescentus* NA1000 reference genome (NCBI Reference Sequence: NC-011916.1) or appropriate reference genomes with mutations at the IBE/YBE/ZBE, using Bowtie 1[71] and the following command: bowtie -m 1 -n 1 −best −strata -p 4 −chunkmbs 512 NA1000-bowtie −sam *.fastq > output.sam. Subsequently, the sequencing coverage at each nucleotide position was computed using BEDTools 2.17.0[72] using the following command: bedtools genomecov -d -ibam output.sorted.bam -g NA1000.fna > coverage_output.txt. When necessary, MACS2 were employed to call peaks[73], for example, using the following command: macs2 callpeak -t./ IHF_exp/output.sorted.bam -c./IHF_control/output.sorted.bam -f BAM -g 4e + 6 −nomodel -n IHFexpvscontrol. Fold-enrichment values, Poisson distribution $-\log_{10}(p \text{ values})$, and false discovery rate $-\log_{10}(q \text{ values})$, as well as visual inspection of both replicates were used to

assess the reproducibility of identified peak. Finally, ChIP-seq profiles were plotted with the x-axis representing genomic positions and the y-axis is the number of reads per base pair per million mapped reads (RPBPM) or number of reads per kb per million mapped reads (RPKPM) using custom R scripts. For the list of ChIP-seq datasets in this study, see Supplementary Data 3. For the statistics of MACS2-detected ChIP-seq peaks, see Supplementary Data 4.

## Co-immunoprecipitation (Co-IP)

*C. crescentus* cells (25 mL) were grown at 28 °C to a stationary phase before cells were harvested by centrifugation. Cell pellets were washed with 1× PBS (pH 7.4), resuspended in 1 mL of lysis buffer (50 mM Tris-HCl pH 8, 150 mM NaCl, 1% Triton X-100, EDTA-free protease inhibitors, 10 mg/mL lysozyme, and 1 μL of Benzonase), and incubated at 37 °C for 20 min. Subsequently, the cell suspension was sonicated on ice using a probe-type sonicator (6 cycles of 15 s with 15 s resting on ice, amplitude setting 8). The lysate was cleared from the cell debris by centrifugation ($17,000 \times g$ for 20 min at 4 °C). 50 μL of this supernatant (the input fraction) was kept for downstream immunoblot analysis. The remaining supernatants were adjusted so that they had the same amount of total protein and mixed with 25 μL anti-FLAG magnetic bead as instructed in the μMACS Epitope Tag Protein Isolation Kit (Miltenyi Biotec). From here, all the subsequent steps were performed according to the instructions from the μMACS kit. The immunoprecipitated proteins (the IP fraction) were eluted using 50 μL of elution buffer (50 mM Tris-HCl pH 6.8, 50 mM DTT, 1% SDS, and 1 mM EDTA).

For western blot analysis related to NusA, NusG and NusE, 10 μg of the input fraction, or 20 μL of the IP fraction for anti-VSVG immunoblots, 10 μL of the IP fraction for anti-ParB, 5 μL of the IP fraction for anti-FLAG immunoblots were loaded on a 4–20% Novex WedgeWell SDS-PAGE gels (Thermo Fisher Scientific). For western blot analysis related to RpoD, 10 μg of the input fraction, 10 μL of the IP fraction for anti-GafY immunoblots, 5 μL of the IP fraction for anti-ParB, 5 μL of the IP fraction for anti-FLAG immunoblots were loaded on a 4–20% Novex WedgeWell SDS-PAGE gels (Thermo Fisher Scientific). Resolved proteins were transferred to polyvinylidene fluoride (PVDF) membranes using the Trans-Blot Turbo Transfer System (BioRad), and the membrane was incubated with a 1:5000 dilution of an anti-VSVG antibody (Sigma-Aldrich, Cat#1970-1 ML), 1:2500 dilution of an anti-FLAG antibody (Merck, Cat# F7425-2MG), or a 1:5000 dilution of an anti-ParB polyclonal antibody (custom antibody, Cambridge Research Biochemicals, UK), or a 1:300 dilution of an anti-GafY polyclonal antibody (custom antibody, Cambridge Research Biochemicals, UK). Subsequently, the membranes were washed twice in a 1× TBS + 0.005% Tween-20 buffer before being incubated in a 1:10,000 dilution of an HRP-conjugated secondary antibody. Blots were imaged using an Amersham Imager 600 (GE Healthcare).

## Growth conditions for IHF-related experiments

*C. crescentus* cells (20 mL) were grown in PYE, in the presence or absence of 0.3% xylose, to stationary phase. Cell pellets were collected from 5 mL cultures for RNA extraction and RT-qPCR analysis. Another 5 mL was collected for immunoblot analysis using anti-CCNA03882 (GtaL, a GTA head-tail connector protein) polyclonal antibody. Another 5 mL was collected for total genomic DNA extraction.

## Genomic DNA extraction

Cell pellets were resuspended in 300 μL cell lysis solution (Qiagen) and lysed by incubation at 50 °C for 10 min. A total of 50 μg of RNaseA was added to the cell lysate, and the lysate was incubated at 37 °C for an hour to remove cellular RNA. Proteins were precipitated by adding 100 μL of a protein precipitation solution (Qiagen). Samples were centrifuged for 5 min at $17,000 \times g$, the resulting supernatant was then mixed with 600 μL of isopropanol, and the tubes were mixed by gentle inversions to precipitate genomic DNA. Genomic DNA was pelleted via

centrifugation at 17,000 × g for 5 min, washed once with 70% ethanol, and resuspended in 200 μL of water.

## Immunoblot to detect GtaL

Cell pellets were resuspended in 300 μL of buffer 1 (20 mM K-HEPES pH 7.9, 50 mM KCl, 10% glycerol and Roche EDTA-free protease inhibitors). Samples were sonicated on ice using a probe-type sonicator (4 cycles of 15 s with 15 s resting on ice, amplitude setting 8). The lysate was cleared from the cell debris by centrifugation at 17,000 × g for 20 min at 4 °C. Bradford assay was used to determine the total protein concentration in each sample so that an equal amount of total proteins were loaded on each well of a 4–20% Novex WedgeWell SDS-PAGE gels (Thermo Fisher Scientific). Resolved proteins were transferred to PVDF membranes using the Trans-Blot Turbo Transfer System (BioRad), and the membrane was incubated with a 1:1000 dilution of an anti-GtaL polyclonal antibodies (custom antibody, Cambridge Research Biochemicals, UK). Subsequently, the membranes were washed twice in a 1 × TBS + 0.005% Tween-20 buffer before being incubated in a 1:10,000 dilution of an HRP-conjugated secondary antibody. Blots were imaged using an Amersham Imager 600 (GE Healthcare).

## β-galactosidase assay

*C. crescentus* cultures (20 mL), inoculated from single colonies, were grown at 28 °C to stationary phase and cooled on ice before β-galactosidase assay. Cultures were diluted fourfold before measuring $OD_{600}$ and 200 μL of diluted cultures were used in the assay. β-galactosidase assays were carried out essentially as follows (see also ref. 74). 200 μL of diluted cultures were added to 800 μL of Z buffer [0.04 M β-mercaptoethanol, 0.06 M $Na_2HPO_4$, 0.04 M $NaH_2PO_4$, 0.01 M KCl, 0.001 M $MgSO_4$] with 30 μL of 0.1% SDS and 60 μL of chloroform. Samples were vortexed and left at room temperature for 30–60 min. 200 μL of 4 mg/mL o-nitrophenyl-β-D-galactoside (ONPG) was added in 10 s interval. The samples were shaken gently to mix and the reactions were stopped by adding 500 μL 1 M $Na_2CO_3$. $OD_{420}$ and $OD_{550}$ were measured for 1 mL of each reaction. 1 mL of Z buffer was used as a reference. β-galactosidase activity was calculated using the formula: $1000 × [(OD_{420} − 1.75 × OD_{550})]/(T × V × OD_{600})$ where $T$ is time of the reaction in minute and $V$ is 0.2 mL. Assays were performed at least twice in duplicates on two different days i.e., four replicates for each experiment.

## Total RNA extraction and quantitative reverse transcriptase PCR (qRT-PCR)

*C. crescentus* cells (20 mL) were grown at 28 °C to stationary phase and cell pellets from 5 mL cultures were collected for total RNA extraction using a Direct-zol RNA miniprep kit (Zymo Research). 10 μg of isolated total RNA was subjected to DnaseI treatment with 20 units of Turbo DnaseI (Invitrogen) for an hour at 37 °C. DnaseI was subsequently removed from total RNA using an RNA clean and concentrator-25 (Zymo Research). Purified RNA isolated from wild-type and *ΔrogA C. crescentus* was sent to Azenta (UK) for RNA-seq. For qRT-PCR, 1 μg of DnaseI-treated total RNA was converted to cDNA using an Invitrogen SuperScript III First-Strand Synthesis SuperMix for qRT-PCR according to the manufacturer's instructions. PCR cycling was performed at 25 °C for 10 min, 42 °C for 120 min, 50 °C for 30 min, 55 °C for 30 min, then 5 min at 85 °C. Following RnaseH treatment, samples were diluted 1:2 with water and 1 μL was used for qRT-PCR using a SYBR® Green JumpStart™ *Taq* ReadyMix™ in a BIORAD CFX96 instrument. Results were analyzed using BIORAD CFX96 software. Transcript quantities for *gafY and gtaT* were determined relative to the amount of *ruvA* transcript, which was selected for being constitutively expressed in the cell growth conditions. Relative expression values were calculated by using the comparative Ct method (ΔΔCt) and were the average of two biological replicates. Error bars represent the relative expression values

calculated from plus or minus one standard deviation from the mean ΔΔCt values. qRT-PCR oligos for quantifying *gafY* transcription are 5'-GCAGCTCGCCATCTACC-3' and 5'-GCAGATCCTCGATCTTGCG-3', for that of *gtaT* (CCNA02880) are 5'-GGCCCTGTACGAGCAAG-3' and 5'-GGCTGTGTTCCAGATCTCC-3', for that of *ruvA* are 5'-ATGGGCGTCGGCTATCT-3' and 5'-CGAGTGAGGAAGCCGTAGA-3'.

## Protein co-overexpression and purification of 6xHis-tagged RpoD, 6xHis-tagged RpoD (domain 3 + 4), and GafY

Plasmid pCOLA-Duet1::6xhis-*rpoD-gafY*, or pCOLA-Duet1::6xhis-*rpoD* (domain 3 + 4) and pET15:: *gafY* were introduced into *E. coli* Rosetta (BL21 DE3) competent cells (Merck) by heat-shock transformation or by electroporation. A 10 mL overnight culture was used to inoculate 1 L of LB medium supplemented with kanamycin and chloramphenicol. Cells were grown at 37 °C with shaking at 210 rpm to an $OD_{600}$ of ~0.4. The culture was then left to cool down to 28 °C before isopropyl-β-D-thiogalactopyranoside (IPTG) was added to a final concentration of 1 mM. The culture was left shaking for an additional 3 h at 28 °C before cells were harvested by centrifugation. Pelleted cells were resuspended in a buffer containing 100 mM Tris-HCl pH 8.0, 300 mM NaCl, 10 mM imidazole, 5% (v/v) glycerol, 1 μL of Benzonase nuclease (Merck), and an EDTA-free protease inhibitor tablet (Merck). The resuspended cells were then lysed by sonication (10 cycles of 15 s with 10 s resting on ice in between each cycle). The cell debris was pelleted by centrifugation at 28,000 × g for 30 min and the supernatant was filtered through a 0.45 μm filter disk. The lysate was then incubated with 2 mL pre-washed HIS-Select Cobalt Affinity Gel (Merck, UK) with rotation for an hour. After an hour, the resin was washed three times with 25 mL buffer A (100 mM Tris-HCl pH 8.0, 300 mM NaCl, 10 mM imidazole, and 5% glycerol). Proteins were eluted from the gel using 2 mL of buffer B (100 mM Tris-HCl pH 8.0, 300 mM NaCl, 500 mM imidazole, and 5% glycerol).

## An in silico screen for protein–protein interactions

A pairwise screen for possible interactions between GafY and *C. crescentus* sigma factors and components of RNA polymerase was conducted using AlphaFold2 Multimer[48] via ColabFold[49]. The confidence metrics (ipTM) for the top model from each pairwise interaction were tabulated, with ipTM > 0.7 indicating a possible protein–protein interaction[75].

## Reporting summary

Further information on research design is available in the Nature Portfolio Reporting Summary linked to this article.

## Data availability

The ChIP-seq data generated in this study have been deposited in the GEO database under accession code GSE247216. All remaining data supporting the findings of this study are available within the paper and its Supplementary Information. Source data are provided with this paper.

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

## Acknowledgements

We thank members of our laboratory, Mark Buttner, Dave Grainger, and Paul Fogg for helpful discussion and comments on this manuscript, and Tom McLean for assistance with an AlphaFold2-based screen. This work is supported by the Royal Society University Fellowship Renewal URF\R\201020, the Lister Institute fellowship, the Wellcome Trust Investigator grant 221776/Z/2/Z (to T.B.K.L.), and the BBSRC-funded Institute Strategic Program Harnessing Biosynthesis for Sustainable Food and Health (HBio) (BB/X01097X/1).

## Author contributions

N.T.T. and T.B.K.L. conceived the project. N.T.T. carried out all experiments and conducted data analysis. T.B.K.L. contributed to ChIP-seq analysis. T.B.K.L. procured funding and supervised the project. N.T.T. and T.B.K.L. wrote the manuscript.

## Competing interests

The authors declare no competing interests.
