## [Peer Review File · Nature Communications]

Control of a gene transfer agent cluster in *Caulobacter crescentus* by transcriptional activation and anti-terminationREVIEWER COMMENTS

Reviewer #1 (Remarks to the Author):

This is a really nice piece of work, showing that transcription of a cluster of *Caulobacter* genes is regulated by a transcription activator, GafY, and an anti-terminator, GafZ. GafY appears to activate multiple target promoters whilst GafZ intervenes just at the polycistronic transcription unit that encodes most of the functional genes for a gene transfer vector 'complex'. Activation by GafY at the promoter of this transcription unit requires help from IHF, a ubiquitous nucleoid-associated protein. The work uses interesting methodologies and is concisely and (mostly) clearly presented. Personally, I greatly enjoyed reading this manuscript, as it raises so many fascinating questions. Having said that, regulation by the combination of an activator and anti-terminator is not uncommon in bacteria, specially with longer transcription units that are subject to horizontal gene transfer. A sceptic might say that the main novelty here lies in the fact that, in the *Rhodobacter capsulatus* iteration of this system, the key regulator, GatA, appears to include fused homologues of *Caulobacter* GafY and GafZ, and the authors of the *Rhodobacter* work may have missed the anti-termination angle, and that this is a somewhat niche observation. My personal view is that there is sufficient in the present manuscript to justify publication in a leading non-specialist journal, but, of course, this is an editorial matter.

However, I do have some suggestions for improving the manuscript:

1. The first sentence of the abstract is out of place here. Suffice it to say that the *Caulobacter crescentus* encodes a set of genes that are responsible for a molecular machine whose biosynthesis needs to be kept in check. If this paper has something to say about the origin of these genes or their relation to phage, please keep it in the Discussion.
2. Line 41 makes no sense to me. Per milliliter of what?
3. Line 94-95 refers to a β -galactosidase reporter, but this will be somewhat confusing to most readers as line 598 refers to lacA encoding a dehydrogenase. Seems to me that the peculiarities of lactose utilisation in *Caulobacter* needs some explanation. See Arellano BH et al. (2010) Identification of a dehydrogenase required for lactose metabolism in *Caulobacter crescentus*. *Appl Environ Microbiol.* 76:3004-14 (of course, maybe it's *E. coli* that is peculiar??)
4. Line 160 and all figures describing ChIP-seq data. Throughout, the authors use a private convention, using α to denote ChIP directed to something (GafY, here, but often an indirect entity such as FLAG). It took me several attempts to grasp the authors' convention, and it needs to be explicitly explained.
5. Line 181, what criteria are used to select this -35 element? What is the *Caulobacter* consensus hexamer? The mutational data pinpointing this element, shown here, is not very strong. Given the suggestion that GafY activates transcript initiation by interacting with housekeeping sigma D3/D4, this is important as we now know that such activation either 'persuades' sigma D4 to dock at a non-consensus sequence or remodels it to produce a new specificity (see Kompaniets D et al (2023) Structure and molecular mechanism of bacterial transcription activation. *Trends Microbiol.*). A complementary strategy might be to change the proposed AATCAT hexamer to the consensus and see if this uncouples expression from GafY. This might be useful later to probe whether GafZ function is enhanced by or independent of GafY.
6. In the diagram in Fig 4a, the scale needs adjusting to give a proportionate spacing between YBE and IBE1. Lines 4 and 5 give a false impression of this spacing. Concerning spacing, nothing is said about the spacing at other promoters such as the gafYZ promoter. I understand completely why the authors don't want to get into this here, but it may just be better to ignore the other targets here, and not 'tempt' the reader. The point is that if the activation mechanism is as suggested, then this spacing is likely to be crucial.
7. Line 225/226 and Discussion. According to Kompaniets D et al (2023), activators that interact with sigma D3/4 bind close to the target promoter -35 element. The published exception is *Caulobacter* CtrA, and there must be others...CtrA can do the trick because it consists of two independently folding domains, one of which binds to DNA targets and the other to sigma. This is clearly not the case for GafY, but there is no discussion of this key point that sets a new paradigm. To my knowledge, the best examples bacterial activators that benefit from IHF-induced DNA bending to contact sigma are EBPs

that interact with sigma54-family factors. Again, this isn't mentioned. At least the text needs to affirm that sigma 54 is not involved here, and mention the evidence that the promoters discussed here are served by the housekeeping sigma factor, and maybe mention the best-currently documented example which is found in Schröder I, et al (1993) Activation of the Escherichia coli nitrate reductase (narGHJI) operon by NarL and Fnr requires integration host factor. J Biol Chem. 268:771-4.

8. Line 228-300. The evidence for GafZ travelling with the RNAP here is compelling. Hence, it seems to me to be confusing to begin this section with GafY ChIP-seq data. The proposed model suggests no role for GafY in antitermination (note that this would be experimentally testable using an improved promoter -35 element that short-circuited the need for GafY in transcript initiation). As the data stand, it looks like the GafY-GafZ interaction (presumably involving GafZ CTD that shows shades of sigma) could account for the skewed GafY ChIP signal, or, alternatively, sigma remaining in the elongation complex. The authors may well already have data on this point, but, to my mind, the present manuscript would read better if the GafY data were moved right to the end. By the way, the bottom row of Fig7a shows that mutating the ZBE makes the GafY ChIP-seq signal symmetric, and this is hardly mentioned.

9. Another point with the antitermination story is that the re-siting of the GtaT translation start creates a long untranslated leader and the data in lines 7-9 argue that GafZ helps the elongation complex to get through this... I recommend giving this attention equal to that of effects at the terminator downstream of gatT.

10. On page 11, in line 338, the ZBE has been 'identified', whilst in line 339, it is described as putative. Sounds like the authors should claim to have identified a putative ZBE (sorry to be so fussy). Following on, the discussion about Q omits to mention that citation 60 explicitly shows how two structurally distinct proteins can end up doing the same job. Seems odd to me that this is ignored here.

Having said all that, this is a great story, but, to my mind, the presentation will be more readable if the authors steered their results to tell the story that they want to get over. Also, I feel that the Discussion and Concluding Remarks sections need a bit more thought and reference to the bigger picture.

Reviewer #2 (Remarks to the Author):

This elegant paper by Tran and colleagues reports a comprehensive series of experiments characterising the transcription regulation of the *C. crescentus* CcGTA gene cluster. The authors use a multidisciplinary approach that demonstrates how IHF, GafY and Z function. I'm particularly fascinated by the proposed anti termination mechanism arising from the interaction of RNAP, known antitermination factors (NusA, G and E) and GafY/Z.

The study is well conceived, the work carried out meticulously, the manuscript is well prepared, results well documented and largely not over interpreted.

Questions and concerns.

1. Are all ChIP-seq profiles statistically significant, in terms of reproducibility, replicates etc? Please provide robust statistical testing of all conclusions based on similarities and differences in occupancy profiles of the components.
2. GafY and Z interact. How come their binding sites / promoter-specificity differs by so much? Explain.
3. Sigma-GafZ interactions potentially rationalise the recruitment of GafZ and Y - but how do the factors interact with RNAP in the elongation complex (assuming that sigma is an initiation factor which dissociates from RNAP proximal to the promoter)? Discuss.
4. What is the underlying molecular mechanism of anti termination, does GafY interact with the contemplate strand DNA, or with the RNA transcript? That's likely the follow-up paper's topic - but at least the authors could provide a short rationale in the discussion section.

Reviewer #3 (Remarks to the Author):

The study focuses on the mechanism of expression of Gene Transfer Agents (GTAs) in *Caulobacter crescentus*. Two proteins, GafY and GafZ, activate the biosynthesis of CcGTA, but their mechanism of action was not known. This study provides evidence that GafY and the Integration Host Factor (IHF) co-activate the CcGTA gene cluster and shows that GafZ is a novel transcription anti-terminator that helps bypass transcription terminators within the CcGTA cluster. The study uncovers a two-tier regulation system coordinating the synthesis of GTA particles in *C. crescentus*. Data is clearly and thoroughly presented throughout, particularly the ChIP data, which made the paper a pleasure to review.

The authors should consider comments below as they revise this manuscript.

- Line 176-179: The authors state that “only ATG number 3 (at position +251) was required for the production of packaged GTA DNA, suggesting that it represents the true *gtaT* start codon”. They examined putative start codons by mutagenizing the putative ATG starts to TGA stop codons. It is not uncommon for proteins to have multiple starts. By mutating their start codons to putative stop codon, the authors not only disrupt the start codon of interest, but also truncate the protein that would originate from earlier start codons. In the case of Figure 3, mutating putative ATG-3 to TGA would result in disruption of a start codon and a truncation of any protein that began translation at putative ATG-1 or putative ATG-2. As such, I think the only claim that the authors can make from the data is that ATG number 3 is the final start codon. In cases where there are multiple starts, you do not see marked reduction of protein unless several start codons are disrupted. It appears that disruption of putative ATG-1 and putative ATG-2 result in reduced GTA DNA accumulation compared to the Δ rogA strain, which could suggest a reduction protein from a removed translation site.
- Figure 1C, 2D, 3B, 7B, S2B, and S6B: It is stated that immunoblots were representative images from at least two biological replicates, but do not state how many replicates were performed for DNA gels. This should be added.
- Line 132-145: It is shown that disruption of *ihfA/B* and/or their binding sites eliminates *gtaT* expression in a Δ rogA background. One model is that *IhfAB* binding enhances *GafY* binding. If disruption of *ihfA/B* reduced *GafY* binding, then one expects a reduction in *gtaT* transcription. Experiments testing this may be outside the scope of this study, but could help clarify the function of *ihfA/B* at the *gtaT* promoter.
- Line 291-292: Deletion of *gafZ* reduced *gtaT* expression (Pgta-153 to +229-lacZ) (Fig 8B), despite *GafY* still binding to promoter region (Fig 7A). Given that this reporter doesn't have the *terGTA*, this would suggest that *GafZ* also plays a role in activation of expression. Some discussion of this possibility would be helpful.
- Line 292-296: Addition of the *terGTA* effectively reduces reporter activity when placed in the *rsaA* promoter. Do the authors predict that addition of a *gafZ* site to the *rsaA* promoter would allow for bypass of the terminator?

Thank you very much for the comments on our manuscript. We are very grateful to all reviewers and the editor for their critical and supportive comments. We have performed additional experiments and revised the manuscript accordingly. We have also re-formatted the manuscript according to the guidelines from Nature Communications. All source data and uncropped gel images have also been uploaded to supplementary files. An article file with revisions made in Track Changes has also been uploaded. Detailed responses to the specific points that reviewers have raised are given below:

REVIEWER COMMENTS

Reviewer #1 (Remarks to the Author):

This is a really nice piece of work, showing that transcription of a cluster of *Caulobacter* genes is regulated by a transcription activator, GafY, and an anti-terminator, GafZ. GafY appears to activate multiple target promoters whilst GafZ intervenes just at the polycistronic transcription unit that encodes most of the functional genes for a gene transfer vector 'complex'. Activation by GafY at the promoter of this transcription unit requires help from IHF, a ubiquitous nucleoid-associated protein. The work uses interesting methodologies and is concisely and (mostly) clearly presented. Personally, I greatly enjoyed reading this manuscript, as it raises so many fascinating questions. Having said that, regulation by the combination of an activator and anti-terminator is not uncommon in bacteria, specially with longer transcription units that are subject to horizontal gene transfer. A sceptic might say that the main novelty here lies in the fact that, in the *Rhodobacter capsulatus* iteration of this system, the key regulator, GatA, appears to include fused homologues of *Caulobacter* GafY and GafZ, and the authors of the *Rhodobacter* work may have missed the anti-termination angle, and that this is a somewhat niche observation. My personal view is that there is sufficient in the present manuscript to justify publication in a leading non-specialist journal, but, of course, this is an editorial matter.

We thank the reviewer for their valuable and encouraging feedback on our manuscript.

However, I do have some suggestions for improving the manuscript:

1. The first sentence of the abstract is out of place here. Suffice it to say that the *Caulobacter crescentus* encodes a set of genes that are responsible for a molecular machine whose biosynthesis needs to be kept in check. If this paper has something to say about the origin of these genes or their relation to phage, please keep it in the Discussion.

We agree and have now removed the phrase "*... but might have been co-opted to perform biological functions for the host bacteria*" from the Abstract.

2. Line 41 makes no sense to me. Per milliliter of what?

We have now rewritten the sentence to clarify the meaning: "*Homologs of GTA core genes are common in abundant environmental organisms, and it has been speculated that $\sim 10^6$ virus-like particles per milliliter of marine water could be GTAs*".

3. Line 94-95 refers to a β -galactosidase reporter, but this will be somewhat confusing to most readers as line 598 refers to lacA encoding a dehydrogenase. Seems to me that the peculiarities of lactose utilisation in *Caulobacter* needs some explanation. See Arellano BH et al. (2010) Identification of a dehydrogenase required for lactose metabolism in *Caulobacter crescentus*. Appl Environ Microbiol. 76:3004-14 (of course, maybe it's *E. coli* that is peculiar??)

We agree, and have now removed the phrase " *β -galactosidase reporter*". We re-wrote the sentence as follows "*To further understand the control of the GTA cluster, we devised a blue-white screen based on a lacA reporter to identify additional factors that might act together with or independently of GafYZ.*" We cited Arellano et al., (2010), further explained the enzymatic activity of LacA, but

added these sentences to the figure legend of Fig. 1a-b to avoid disrupting the flow of the main text. Specifically, we wrote, “*C. crescentus* forms blue colonies on agar media supplemented with a chromogenic lactose analog (X-gal) owing to the presence of a membrane-bound dehydrogenase LacA that is necessary for converting lactose/lactose analog into molecules that can be imported into the cytoplasm for subsequent hydrolysis (Arellano et al., 2010), while Δ lacA colonies are white on such media (Fig. 1b).”

4. Line 160 and all figures describing ChIP-seq data. Throughout, the authors use a private convention, using α to denote ChIP directed to something (GafY, here, but often an indirect entity such as FLAG). It took me several attempts to grasp the authors' convention, and it needs to be explicitly explained.

We have now modified the text and figures throughout according to the reviewer's suggestion. Specifically, we use “anti-FLAG or anti-VSVG” instead of “ α -FLAG or α -VSVG” to denote the antibodies used in ChIP-seq experiments.

5. Line 181, what criteria are used to select this -35 element? What is the *Caulobacter* consensus hexamer? The mutational data pinpointing this element, shown here, is not very strong. Given the suggestion that GafY activates transcript initiation by interacting with housekeeping sigma D3/D4, this is important as we now know that such activation either ‘persuades’ sigma D4 to dock at a non-consensus sequence or remodels it to produce a new specificity (see Kompaniets D et al (2023) Structure and molecular mechanism of bacterial transcription activation. Trends Microbiol.). A complementary strategy might be to change the proposed AATCAT hexamer to the consensus and see if this uncouples expression from GafY. This might be useful later to probe whether GafZ function is enhanced by or independent of GafY.

The sequences for core elements (-10 and -35) of *C. crescentus* sigma 70-dependent promoters were identified in Malakooti et al., 1995 (J Malakooti 1, S P Wang, B Ely. A consensus promoter sequence for *Caulobacter crescentus* genes involved in biosynthetic and housekeeping functions. J Bacteriol. 1995 Aug;177(15):4372-6. doi: 10.1128/jb.177.15.4372-4376.1995.). Malakooti identified the consensus -35 sequence to be TTGACG, which is similar to the *E. coli* -35 one. Based on this publication and the identified TSS of *gtaT*, we identified a putative -10 and a degenerate putative -35 region (Fig. 3).

Based on the reviewer's suggestion, we have now changed the proposed AATCATc of the *PgtaT* (-153 to +229)-*lacZ* construct to the consensus (aTTGTCTG, the -35 region of a highly expressed sigma 70-dependent *rsaA* gene in *C. crescentus*). The β -galactosidase activity from this chimeric promoter was reduced ~sixfold compared to the native promoter, however, it is now “ON” regardless of the presence or absence of GafYZ (Supplementary Fig. 4). We, therefore, reason that GafY activates transcription most likely by assisting sigma D4 to bind a non-consensus sequence rather than remodeling it to produce a new specificity. We have now added the new data to Supplementary Fig. 4, expanded the Discussion and cited Kompaniets D et al., 2023.

6. In the diagram in Fig 4a, the scale needs adjusting to give a proportionate spacing between YBE and IBE1.

Done.

Lines 4 and 5 give a false impression of this spacing. Concerning spacing, nothing is said about the spacing at other promoters such as the *gafYZ* promoter. I understand completely why the authors don't want to get into this here, but it may just be better to ignore the other targets here, and not ‘tempt’ the reader. The point is that if the activation mechanism is as suggested, then this spacing is likely to be crucial.

Based on the reviewer's feedback, we have performed additional experiments and showed that insertion of 6 or 10 additional nucleotides in between YBE and IBE1 (red arrows, see the figure below) eliminated the promoter activity (figure below). However, we wish to reserve these data and perform more insertions at various locations between YBE and IBE to explore the possibility of DNA looping more rigorously in a future manuscript. I hope the reviewer will agree, however, we will add these data to the current manuscript if the reviewer insists. N.B. we have expanded the Discussion to mention the potential importance of spacing for transcriptional activation.

7. Line 225/226 and Discussion. According to Kompaniets D et al (2023), activators that interact with sigma D3/4 bind close to the target promoter -35 element. The published exception is *Caulobacter* CtrA, and there must be others...CtrA can do the trick because it consists of two independently folding domains, one of which binds to DNA targets and the other to sigma. (We assume the reviewer meant *Caulobacter* GcrA rather than CtrA? PMID: 36715319 DOI: 10.1093/nar/gkad016; PMID: 26545812 DOI: 10.1101/gad.270660.115). This is clearly not the case for GafY, but there is no discussion of this key point that sets a new paradigm. To my knowledge, the best examples bacterial activators that benefit from IHF-induced DNA bending to contact sigma are EBPs that interact with sigma54-family factors. Again, this isn't mentioned. At least the text needs to affirm that sigma 54 is not involved here, and mention the evidence that the promoters discussed here are served by the housekeeping sigma factor, and maybe mention the best-currently documented example which is found in Schröder I, et al (1993) Activation of the *Escherichia coli* nitrate reductase (*narGHJI*) operon by NarL and Fnr requires integration host factor. *J Biol Chem.*268:771-4.

We thank the reviewer for these excellent points, especially for bringing the work by Schröder I et al., (1993) to our attention. We have now substantially expanded the Discussion according to the reviewer's suggestions.

During the revision of this manuscript, we constructed FLAG-tagged versions of *rpoN* to perform ChIP-qPCR/seq to test if sigma factor 54 is enriched at GTA promoters. However, none of the *flag-rpoN* or *rpoN-flag* were functional as judged by our inability to immunoprecipitate DNA from known sigma 54-dependent promoters in *C. crescentus*. Given the time constraint, we could not raise polyclonal antibodies against *C. crescentus* RpoN to test further. We, however, have inspected the -12 and -24 regions of GTA promoters and together with the multiple evidence from Fig. 3c,

Supplementary Fig. 4, and Fig. 5, it is most likely that sigma factor 70 (rather than sigma factor 54) is involved in the transcription of the GTA cluster. We have now emphasized this in the Discussion.

8. Line 228-300. The evidence for GafZ travelling with the RNAP here is compelling. Hence, it seems to me to be confusing to begin this section with GafY ChIP-seq data. The proposed model suggests no role for GafY in antitermination (note that this would be experimentally testable using an improved promoter -35 element that short-circuited the need for GafY in transcript initiation). As the data stand, it looks like the GafY-GafZ interaction (presumably involving GafZ CTD that shows shades of sigma) could account for the skewed GafY ChIP signal, or, alternatively, sigma remaining in the elongation complex. The authors may well already have data on this point, but, to my mind, the present manuscript would read better if the GafY data were moved right to the end. By the way, the bottom row of Fig7a shows that mutating the ZBE makes the GafY ChIP-seq signal symmetric, and this is hardly mentioned.

We agree and have now rewritten this section to start with a description of GafZ ChIP-seq instead.

We have also mentioned in the Results that mutating the ZBE makes the GafY ChIP-seq peak symmetric. We wrote, "*However, GafY was no longer enriched in the coding region of gtaT and the main GTA cluster (Fig. 7b), possibly because GafZ (and thereby the GafZ-GafY complex) failed to form a transcription elongation complex with RNAP when ZBE was mutated.*"

On the point about "*The proposed model suggests no role for GafY in antitermination (note that this would be experimentally testable using an improved promoter -35 element that short-circuited the need for GafY in transcript initiation)*", we wish to construct more synthetic promoters (and controls) beyond the one we have already constructed and mentioned in point 5 (Supplementary Fig. 4) to better uncouple GafY activity from that of GafZ. We reason that this is more suitable for a follow-up manuscript, and hope the reviewer will agree.

9. Another point with the antitermination story is that the re-siting of the GtaT translation start creates a long untranslated leader and the data in lines 7-9 argue that GafZ helps the elongation complex to get through this... I recommend giving this attention equal to that of effects at the terminator downstream of gtaT.

We agree and have now mentioned in the Discussion the possibility that GafZ and its anti-termination activity assist the elongation complex to travel through a long untranslated leader region.

10. On page 11, in line 338, the ZBE has been 'identified', whilst in line 339, it is described as putative. Sounds like the authors should claim to have identified a putative ZBE (sorry to be so fussy).

Corrected

Following on, the discussion about Q omits to mention that citation 60 explicitly shows how two structurally distinct proteins can end up doing the same job. Seems odd to me that this is ignored here.

This is an important point, we have now explicitly mentioned how two structurally distinct proteins can have the same anti-termination functions in the Discussion. Specifically, we wrote "*While AlphaFold-predicted structures of GafZ show no sequence or structural similarity to these known anti-terminators, it has been shown recently that Q protein of bacteriophage 21 (Q21) and λ Q, despite sharing no sequence similarity, both form a nozzle that narrows and extends the RNAP RNA-exit channel to prevent the formation of RNA hairpin⁶². Future works, especially solving a cryo-EM structure of GafYZ-RNAP-DNA holo-enzyme complex, will hope to determine how GafZ mechanistically modifies RNAP to mediate transcription anti-termination and whether GafZ shares the same mechanism as AlpA, Q21, and λ Q proteins.*"

Having said all that, this is a great story, but, to my mind, the presentation will be more readable if the authors steered their results to tell the story that they want to get over. Also, I feel that the Discussion and Concluding Remarks sections need a bit more thought and reference to the bigger picture.

We agree and have incorporated the reviewer's suggestions.

Reviewer #2 (Remarks to the Author):

This elegant paper by Tran and colleagues reports a comprehensive series of experiments characterising the transcription regulation of the *C. crescentus* CcGTA gene cluster. The authors use a multidisciplinary approach that demonstrates how IHF, GafY and Z function. I'm particularly fascinated by the proposed anti termination mechanism arising from the interaction of RNAP, known antitermination factors (NusA, G and E) and GafY/Z.

The study is well conceived, the work carried out meticulously, the manuscript is well prepared, results well documented and largely not over interpreted.

Questions and concerns.

1. Are all ChIP-seq profiles statistically significant, in terms of reproducibility, replicates etc? Please provide robust statistical testing of all conclusions based on similarities and differences in occupancy profiles of the components.

We followed the guidelines by ENCODE for performing and reporting ChIP-seq peaks and used MACS2 to detect and test if peaks are statistically significant according to Poisson distribution p-value and false discovery rate q value (p and q values $< 10^{-5}$ or $-\log_{10}(p \text{ or } q \text{ values}) > 5$ are considered to be significant ChIP-seq peaks).

ChIP-seq peaks reported in the manuscript and figures are present and highly significant in both replicates as judged by these p and q values ($-\log_{10}(p \text{ or } q \text{ values})$ of peaks > 200). We have (i) added a new Supplementary Table to report these p and q values, and (ii) we have added p and q values of peaks in both replicates to the figure legends and updated the Materials and Methods as well.

We have also added the Pearson's correlation values of anti-FLAG-GafZ ChIP-seq profile (from 3010 kb to 3030 kb) vs. anti-RpoC-FLAG, anti-VSVG-NusA/G/E ChIP-seq profiles (Fig. 6a) ($r = 0.64, 0.93, 0.95,$ and $0.92,$ respectively, $p < 10^{-16}$) to indicate that GafZ likely co-travels with RpoC and NusAGE (Fig. 6).

Lastly, we added in Supplementary Table 4 the number of mappable reads used to report ChIP-seq profiles (the minimal number of 50-bp reads per sample is 2.3 million, and on average, there are 7 million 50-bp reads per sample. The *Caulobacter* genome size is ~4Mb, so this represents a high coverage i.e. ~25 fold).

2. GafY and Z interact. How come their binding sites / promoter-specificity differs by so much? Explain.

GafY and GafZ were shown to interact (Gozzi *et al.*, 2021 PLoS Biology) but it is not known how stable or transient this interaction is and whether formaldehyde, which was employed to fix cells for ChIP-seq, captured protein-protein interaction as efficiently as protein-DNA interaction. We note that both GafY and GafZ have predicted DNA-binding domains of their own. However, only the promoter of the main GTA cluster (*Pgta7*) has a dedicated GafZ-binding site (ZBE) which is not present elsewhere on the chromosome. We speculate that this additional GafZ-ZBE DNA interaction at the

PgtA contributes to the promoter specificity there. We have expanded the Discussion to discuss this point.

3. Sigma-GafZ (we assume the reviewer meant Sigma-GafY interaction here) interactions potentially rationalise the recruitment of GafZ and Y - but how do the factors interact with RNAP in the elongation complex (assuming that sigma is an initiation factor which dissociates from RNAP proximal to the promoter)? Discuss.

We reason that GafZ binding to ZBE at the promoter of the main GTA cluster (*PgtA*) enables GafZ to load onto RNAP to enhance the processivity of RNAP; this speculation is based on functional similarity to well-characterized anti-termination factors such as Q or AlpA (see Discussion). We are collaborating with a cryo-EM laboratory to solve the structure of the GafY-GafZ-RNAP holo-enzyme complex. Without such structure, it is unclear how GafZ and GafY (via the GafY-Z interaction) interact with RNAP in the elongation complex. We refrain from speculating due to the lack of structural data, but hope to shed more light on how GafYZ works mechanistically in a future study. Specifically, we wrote “*The ChIP-seq profiles of C. crescentus GafZ are also reminiscent of the binding patterns of RNAP and transcription anti-terminators such as P. aeruginosa AlpA, E. coli RfaH, and λQ^{57,59–61}. The λQ binding element (QBE) and the AlpA binding element (ABE) help the direct loading of Q and AlpA onto RNAP^{57,58,62–65}, and it seems likely that the putative GafZ binding element (ZBE) in the GTA promoter may help loading of GafZ onto RNAP in a similar way, allowing RNAP to bypass the long untranslated region and downstream transcriptional terminators (Fig. 9b). The cryo-EM structures of λQ- and AlpA-bound RNAP have been solved, showing that these anti-terminators form a molecular nozzle near the RNA exit channel of RNAP to prevent the formation of terminator hairpin structures that would otherwise form in the nascent RNA and thereby impede or stop transcription elongation^{58,62,66}. While AlphaFold-predicted structures of GafZ show no sequence or structural similarity to these known anti-terminators, it has been shown recently that Q protein of bacteriophage 21 (Q21) and λQ, despite sharing no sequence similarity, both form a nozzle that narrows and extends the RNAP RNA-exit channel to prevent the formation of RNA hairpin⁶². Future works, especially solving a cryo-EM structure of GafYZ-RNAP-DNA holo-enzyme complex, will help to determine how GafZ mechanistically modifies RNAP to mediate transcription anti-termination and whether GafZ shares the same mechanism as AlpA, Q21, and λQ proteins.*”

4. What is the underlying molecular mechanism of anti-termination, does GafY interact with the contemplete strand DNA, or with the RNA transcript? That’s likely the follow-up paper’s topic - but at least the authors could provide a short rationale in the discussion section.

For the same reason as explained in point 3, we hope to shed more light on how GafZ mechanistically anti-terminates in a future study. While we refrain from speculating too far, we have expanded the Discussion to discuss a possible mechanism of anti-termination (please see point 3 and Discussion)

Reviewer #3 (Remarks to the Author):

The study focuses on the mechanism of expression of Gene Transfer Agents (GTAs) in *Caulobacter crescentus*. Two proteins, GafY and GafZ, activate the biosynthesis of CcGTA, but their mechanism of action was not known. This study provides evidence that GafY and the Integration Host Factor (IHF) co-activate the CcGTA gene cluster and shows that GafZ is a novel transcription anti-terminator that helps bypass transcription terminators within the CcGTA cluster. The study uncovers a two-tier regulation system coordinating the synthesis of GTA particles in *C. crescentus*. Data is clearly and thoroughly presented throughout, particularly the ChIP data, which made the paper a pleasure to review.

The authors should consider comments below as they revise this manuscript.

- Line 176-179: The authors state that “only ATG number 3 (at position +251) was required for the production of packaged GTA DNA, suggesting that it represents the true *gtaT* start codon”. They examined putative start codons by mutagenizing the putative ATG starts to TGA stop codons. It is not uncommon for proteins to have multiple starts. By mutating their start codons to putative stop codon, the authors not only disrupt the start codon of interest, but also truncate the protein that would originate from earlier start codons. In the case of Figure 3, mutating putative ATG-3 to TGA would result in disruption of a start codon and a truncation of any protein that began translation at putative ATG-1 or putative ATG-2. As such, I think the only claim that the authors can make from the data is that ATG number 3 is the final start codon. In cases where there are multiple starts, you do not see marked reduction of protein unless several start codons are disrupted. It appears that disruption of putative ATG-1 and putative ATG-2 result in reduced GTA DNA accumulation compared to the Δ rogA strain, which could suggest a reduction protein from a removed translation site.

We thank the reviewer for pointing this out and have now referred the ATG3 as the final start codon rather than the correct start codon.

- Figure 1C, 2D, 3B, 7B, S2B, and S6B: It is stated that immunoblots were representative images from at least two biological replicates, but do not state how many replicates were performed for DNA gels. This should be added.

We have now added to the figure legends to clarify that DNA gels were representative images from at least two biological replicates.

- Line 132-145: It is shown that disruption of *ihfA/B* and/or their binding sites eliminates *gtaT* expression in a Δ rogA background. One model is that IhfAB binding enhances GafY binding. If disruption of *ihfA/B* reduced GafY binding, then one expects a reduction in *gtaT* transcription. Experiments testing this may be outside the scope of this study, but could help clarify the function of *ihfA/B* at the *gtaT* promoter.

Figure 2c (right panel) showed that deleting *ihfA* or *ihfB* from Δ rogA cells completely eliminated (*not just reduced*) *gtaT* transcription, as judged by qRT-PCR. Due to this observation, we reason that the model where GafY (without IHF) can activate some transcription and IHF enhances GafY binding to further activate transcription is less likely.

- Line 291-292: Deletion of *gafZ* reduced *gtaT* expression ($P_{gta-153}$ to +229-*lacZ*) (Fig 8B), despite GafY still binding to promoter region (Fig 7A). Given that this reporter doesn't have the *terGTA*, this would suggest that GafZ also plays a role in activation of expression. Some discussion of this possibility would be helpful.

This is an excellent point. Reviewer 1 (in point 9) also pointed out the possibility that GafZ and its anti-termination activity assist the elongation complex to travel through a long untranslated leader region even though there is no *terGTA* in the $P_{gta-153}$ to +229-*lacZ*. We have added this to the Discussion.

- Line 292-296: Addition of the *terGTA* effectively reduces reporter activity when placed in the *rsaA* promoter. Do the authors predict that addition of a *gafZ* site to the *rsaA* promoter would allow for bypass of the terminator?

Based on the reviewer's suggestions, we replaced the spacer between the -35 and -10 of P_{rsaA} with the GafZ-binding element (ZBE) to construct a chimeric promoter (P_{rsaA} + ZBE) (see figure below). We observed that inserting ZBE did not affect transcription from P_{rsaA} . However, when *terGTA* was inserted downstream of such chimeric promoter i.e. P_{rsaA} + ZBE + *terGTA*, β -galactosidase activity was abolished (see figure below). Our data seemingly suggested that the addition of ZBE to the P_{rsaA} did not allow the bypass of the terminator. There is, however, a caveat to this experiment because

there was no GafY-binding element (YBE) being added and we do not yet fully understand how GafY and GafZ might work together to confer anti-termination. We feel that more synthetic promoters (and controls) are required to answer the reviewer's question satisfactorily. We hope that the reviewer will agree that this is more suitable for a follow-up study.

REVIEWERS' COMMENTS

Reviewer #1 (Remarks to the Author):

The revised version of this paper addresses the points made by the reviewers so it is now good to go I picked up just a few very minor points:

1. the authors use the adverb 'mechanistically' several times in the context of understanding transcription e.g, in the abstract & line 353. In my opinion, its use is superfluous
2. line 324... the only other (not another)
3. line 358... seemingly at odds with... (not odd)
4. line 388, point out that IHF is a globally-acting factor#

Concerning Author Response to Reviewer 1. Point 6, this reviewer is not insisting. Point 7, apologies for misnaming TF. Point 8, this reviewer agrees about follow-up manuscript

Reviewer #2 (Remarks to the Author):

Excellent revision, all systems go as far as I'm concerned!

Reviewer #3 (Remarks to the Author):

The authors have appropriately responded to my initial review. I have no additional concerns.

Thank you very much for the comments on our manuscript. We are very grateful to all reviewers and the editor for their critical and supportive comments. We have also re-formatted the manuscript according to the guidelines from Nature Communications. All source data and uncropped gel images have also been uploaded to supplementary files. Detailed responses to the specific points that reviewers have raised are given below:

Reviewer #1 (Remarks to the Author):

The revised version of this paper addresses the points made by the reviewers so it is now good to go

I picked up just a few very minor points:

1. the authors use the adverb 'mechanistically' several times in the context of understanding transcription e.g, in the abstract & line 353. In my opinion, its use is superfluous

We have now removed these adverbs. Thank you for the suggestion to improve the manuscript.

2. line 324... the only other (not another)

Corrected

3. line 358... seemingly at odds with... (not odd)

Corrected

4. line 388, point out that IHF is a globally-acting factor#

Done

Concerning Author Response to Reviewer 1. Point 6, this reviewer is not insisting. Point 7, apologies for misnaming TF. Point 8, this reviewer agrees about follow-up manuscript

Thank you

Reviewer #2 (Remarks to the Author):

Excellent revision, all systems go as far as I'm concerned!

Thank you

Reviewer #3 (Remarks to the Author):

The authors have appropriately responded to my initial review. I have no additional concerns.

Thank you